# CONTRADIFF: PLANNING TOWARDS HIGH RETURN STATES VIA CONTRASTIVE LEARNING

**Yixiang Shan, Zhengbang Zhu, Ting Long**[*]**, Qifan Liang, Yi Chang**[*]**, Weinan Zhang, Liang Yin**
[1] Jilin University, [2] Shanghai Jiao Tong University
{shanyx22, liangqf23}@mails.jlu.edu.cn
{zhengbangzhu, wnzhang}@sjtu.edu.cn
{longting, yichang}@jlu.edu.cn
yinla@apex.sjtu.edu.cn

## ABSTRACT

The performance of offline reinforcement learning (RL) is sensitive to the proportion of high-return trajectories in the offline dataset. However, in many simulation environments and real-world scenarios, there are large ratios of low-return trajectories rather than high-return trajectories, which makes learning an efficient policy challenging. In this paper, we propose a method called Contrastive Diffuser (ContraDiff) to make full use of low-return trajectories and improve the performance of offline RL algorithms. Specifically, ContraDiff groups the states of trajectories in the offline dataset into high-return states and low-return states and treats them as positive and negative samples correspondingly. Then, it designs a contrastive mechanism to pull the planned trajectory of an agent toward high-return states and push them away from low-return states. Through the contrast mechanism, trajectories with low returns can serve as negative examples for policy learning, guiding the agent to avoid areas associated with low returns and achieve better performance. Through the contrast mechanism, trajectories with low returns provide a "counteracting force" guides the agent to avoid areas associated with low returns and achieve better performance. Experiments on 27 sub-optimal datasets demonstrate the effectiveness of our proposed method. Our code is publicly available at https://github.com/Looomo/contradiff.

## 1 INTRODUCTION

Offline reinforcement learning (offline RL) (Levine et al., 2020; Prudencio et al., 2023) is a significant branch of reinforcement learning, where an agent is trained on pre-collected offline datasets and is evaluated online. Since offline RL avoids potential risks from interacting with the environment during policy learning, it has broad applications in numerous real-world scenarios, like commercial recommendation (Xiao & Wang, 2021), health care (Fatemi et al., 2022), dialog systems (Jaques et al., 2020), and autonomous driving (Shi et al., 2021).

However, the performance of offline RL methods highly depends on the proportion of the high-return trajectories in the offline dataset. When the dataset contains a large proportion of high-return trajectories, as is presented in Figure 1(b), offline RL methods can easily learn the pattern of high-return trajectories such that they can achieve excellent performance when interacting with the environment. In contrast, when the dataset has a limited number of high-return trajectories, as is presented in Figure 1 (c), offline RL methods struggle to learn a good pattern from the dataset to achieve high returns(Hong et al., 2023b). Unfortunately, the issue of limited high-return trajectories commonly exists in both simulation environments (*e.g.*, Maze2d) and real-world scenarios (*e.g.*, robotics control and medical diagnosis). As it is illustrated in Figure 1(a), we visualize the probability density of trajectories' returns in Maze2d. We can observe that the number of high-return trajectories is much limited.

To address this issue, previous works (Hong et al., 2023b;a) typically reweight the importance of samples, assigning higher weights to high-return trajectories (as shown in Figure 1(d)). However, these methods essentially focus on learning the patterns of high-return trajectories, encouraging the agent to replicate actions that lead to higher rewards during interactions with the environment. Nevertheless, if the agent starts in a low-return region or unexpectedly falls into a low-return region

---

[*]Corresponding Authors.

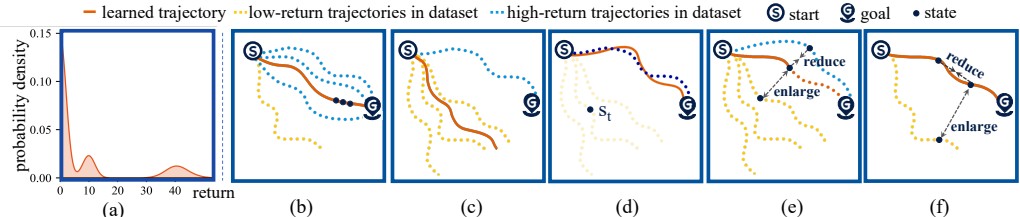

Figure 1: (a) The probability density of trajectories' returns in Maze2d; (b) The learned trajectory when high-return trajectories are abundant; (c) The learned trajectory when the number of high-return trajectories is limited; (d) Previous address high-return trajectories limited issue by increasing the weight of high-return trajectories; (e) The example of our solution; (f) The contrastive learning applied by previous RL models.

during the interaction, as illustrated by state $s_t$ in Figure 1(d), ignoring low-return areas during training makes it hard for these methods to escape from $s_t$, since there's no existing path from $s_t$ to goal. Therefore, in the case of sparse high-return trajectories, focusing solely on learning from high-return trajectories is insufficient.

In fact, states in high-return trajectories indicate potential areas where high returns can be obtained, while states in low-return trajectories indicate the potential regions where agents might encounter low returns. Therefore, one promising way to fully utilize both low-return and high-return trajectories in an offline dataset, is using high-return trajectories as examples to attract the agent to stay as close as possible, while using states from low-return trajectories to indicate potential low-return regions, guiding the agent to stay away from, as shown in Figure 1(e).

However, there are no mature techniques to pull the states of a trajectory toward high-return states and push them away from low-return states, to the best of our knowledge. Fortunately, there is an analogous case: contrastive learning, which aims to bring a given sample close to positive (*i.e.*, similar) samples and far from negative (*i.e.*, dissimilar) samples (Xiao et al., 2020; Tian et al., 2020; Wang & Qi, 2022; Khosla et al., 2020). Inspired by that, to better use the valuable insights implicated in low-return trajectories, we propose to treat states with high return in trajectories of offline dataset as positive samples and those with low return as negative samples, and leverage contrastive learning to pull the states toward high-return states and push them away from low-return states, as Figure 1(e) illustrates. It is worth noting that, unlike previous works (Qiu et al., 2022; Laskin et al., 2020; Yuan & Lu, 2022; Agarwal et al., 2020), which apply contrastive learning to constrain the states of the same trajectory to similar representations and the states of different trajectories to dissimilar representations, as is illustrated in Figure 1(f), we aim to **use contrastive learning to constrain policy toward high-return states and away from low-return states**, as is illustrated in Figure 1(d). Furthermore, the criteria for distinguishing positive and negative samples here are based on the returns rather than the labels.

Through the contrast mechanism, trajectories with low returns can serve as counteracting force for policy learning, guiding the agent to avoid areas associated with low returns. Additionally, with the guidance of high-return states, the agent ultimately achieves high returns. However, ordinary states are feedback from the environment rather than generated by the model, applying contrastive mechanisms to these states produces no gradient for policy optimization. Considering some diffusion-based RL methods generate subsequent trajectories for planning (Janner et al., 2022; Ajay et al., 2023), in which abundant states are generated by policy model, we build our constrastive mechanism on those diffusion-based RL methods and propose a method called **Contrastive Diffuser** (**ContraDiff**). Specifically, we first group the states of the trajectories in the offline dataset into high-return states and low-return states. Then, we learn a diffusion-based trajectory generation model to generate the subsequent trajectories, and apply a contrastive mechanism to constrain the states of the generated trajectories by pulling them toward the high-return states and pushing them away from the low-return states in the offline dataset. With the contrastive mechanism constrained states for planning, the agent makes decisions towards the high-return states. Experiment results on 27 sub-optimal datasets demonstrate the superior performance of ContraDiff.

In summary, our contributions are: (i) We propose a method called ContraDiff, which takes the advantage of low-return trajectories by pulling the states in trajectories toward to high-return states and pushing them away from low-return states. (ii) We perform contrastive learning to constrain the states in the agent's planned trajectory and enhance the policy learning. To the best of our knowledge, our work is the first which apply contrastive learning to directly improve the policy learning. (iii) Experimental comparison with SOTA methods on 27 sub-optimal datasets and thorough further investigations demonstrate the effectiveness of ContraDiff.

## 2 PRELIMINARIES

### 2.1 DENOISING PROBABILISTIC MODELS

Denoising Diffusion Probabilistic Models (Diffusion Models) (Sohl-Dickstein et al., 2015; Song et al.; Ho et al., 2020) are a group of generative models, which generate samples by denoising from Gaussian noise. A diffusion model is composed of a forward process and a backward process. Given the original data $x \sim q(x)$, the forward process transfers $x$ into Gaussian noise by gradually adding noise, *i.e.*, $q(x^i|x^{i-1}) = \mathcal{N}(x^i; \sqrt{1 - \beta^i}x^{i-1}, \beta^i I)$, in which $I$ is an identity matrix, $\beta^i$ is the noise schedule measuring the proportion of noise added at each step, $x^0 := x$ is a sample from the offline dataset, $x^1, x^2, ...$ are the latents of diffusion. The backward process recovers $x$ by gradually removing the noise at each step, which is formulated with a Gaussian distribution (Feller, 1949) parameterized by $\theta$, *i.e.*, $p_\theta(x^{i-1}|x^i) = \mathcal{N}(\mu_\theta(x^i, i), \Sigma_\theta(x^i, i))$, where $\mu_\theta(x^i, i) = \frac{\sqrt{\alpha^i}(1-\bar{\alpha}^i)}{1-\bar{\alpha}^{i-1}}x^i + \frac{\sqrt{\bar{\alpha}^{i-1}}\beta^i}{1-\bar{\alpha}^i}\psi_\theta(x^i, i)$, $\bar{\alpha}^i = \prod_{j=1}^{i}(1 - \beta^i)$ and $\psi_\theta(\cdot, \cdot)$ is a model to reconstruct $x$. The objective function can be formulated as follows if we fix $\Sigma_\theta(x^i, t) = \beta_i I$ (Ho et al., 2020):

$$\mathcal{L} = \mathbb{E}_{x^0, i\sim[1,N]} \left[ \|x^0 - \psi_\theta(x^i, i)\|^2 \right]. \tag{1}$$

### 2.2 CONTRASTIVE LEARNING

Contrastive learning (Schroff et al., 2015; Sohn, 2016; Khosla et al., 2020; Yeh et al., 2022; Oord et al., 2018) is a class of self-supervised learning methods which aim at pulling similar samples together and pushing dissimilar samples away from each other. Specifically, given a sample $x$ and a similarity measure, the positive set $\mathcal{S}^+$ is defined as the collection of samples similar to $x$, while the negative set $\mathcal{S}^-$ is defined as the collection of samples dissimilar to $x$. Contrastive learning minimizes the distance of between $x$ and $\mathcal{S}^+$, and maximizes the distance of $x$ and $\mathcal{S}^-$. That is, for each sample $x$, select a positive sample $x^+ \in \mathcal{S}^+$ and negative samples $x^- \in \mathcal{S}^-$, the learning loss is:

$$\mathcal{L} = -\log \left[ \frac{\exp(\text{sim}(f(x), f(x^+)))}{\exp(\text{sim}(f(x), f(x^+))) + \sum_{x^-\in\mathcal{S}^-} \exp(\text{sim}(f(x), f(x^-)))} \right], \tag{2}$$

where $f(\cdot)$ is the function to map samples to a latent space and $\text{sim}(\cdot, \cdot)$ is the similarity measure.

### 2.3 OFFLINE RL PROBLEM DEFINITION

Considering a system composed of three parts: policy, agent, and environment. The environment in RL is usually formulated as a Markov Decision Process (MDP) (Sutton & Barto, 2018) $\mathcal{M} = \{\mathcal{S}, \mathcal{A}, \mathcal{P}, r, \gamma\}$, where $\mathcal{S}$ is the state space, $\mathcal{A}$ is the action space, $\mathcal{P}(s'|s, a)$ is the transition function, $\gamma$ represents the discount factor, $r$ is the instant reward of each step. At each step $t$, the agent responds to the state of environment $s_t$ by action $a_t$ according to policy $\pi_\theta$ parameterized by $\theta$, and gets an instant return $r_t$. The interaction history is formulated as a trajectory $\tau = \{(s_t, a_t, r_t)|t \geq 0\}$. In this paper, we define the cumulative discounted reward from step $t$ as $v_t = \sum_{i\geq t} \gamma^i r_i$ and call it the return of $s_t$.

We focus on the offline RL setting in this paper. Given an offline dataset $\mathcal{D} \triangleq \{(s_t, a_t, r_t, s_{t+1})|t \geq 0\}$ consisting of transition tuples, and defining the return of trajectory $\tau$ as $R(\tau) \triangleq \sum_{t\geq 0} \gamma^t r_t$, our goal is learning $\pi_\theta$ to maximize the expected return without directly interacting with the environment, *i.e.*,

$$\pi_\theta = \arg\max_\theta \mathbb{E}_{\tau\sim\pi_\theta}[R(\tau)]. \tag{3}$$

## 3 METHODOLOGY

As we discussed previously, the performance of offline RL methods is suppressed when the number of high-return trajectories is limited. Previous methods of increasing the weights of high-return samples fail to utilize the low-return samples and are ineffective in addressing situations where the agent gets stuck in regions with low returns. To address this challenge, we propose a method called **Constrastive Diffuser (ContraDiff)**, which introduces a contrastive mechanism to make full use of low-return trajectories and enhance the performance by constraining the states of the agent's trajectory towards high-return states and away from low-return states. As is illustrated in Figure 2, Our ContraDiff is composed of two modules: (1) the Planning Module, which aims to generate subsequent trajectories; (2) the Contrastive Module, which is designed to constrain the states in generated trajectories within the high-return areas and away from low-return areas.

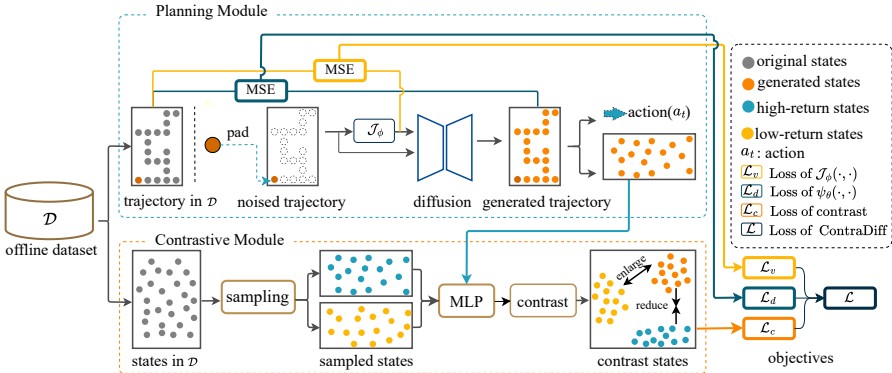

Figure 2: The overall framework of ContraDiff. ContraDiff is composed of two modules: the Planning Module and the Contrastive Module. The Planning Module is designed to generate the subsequent trajectories, and the Contrastive Module is designed to pull the states in the generated trajectories toward the high-return states and push them away from the low-return states during the training phase.

## 3.1 PLANNING MODULE

We designed the Planning Module as an $N$-step Denoising Diffusion Probabilistic Model (Sohl-Dickstein et al., 2015; Song et al.; Ho et al., 2020). Given a state $s_t$ at RL step $t$, the Planning Module first generates a $H$-length subsequent trajectory $\hat{\tau}_t^0$ by alternately denoising generated trajectories and estimating trajectory returns, and then extract the action to be executed from $\hat{\tau}_t^0$, as is illustrated in Figure 2. Specifically, we first sample $\hat{\tau}_t^N$ from $\mathcal{N}(\mathbf{0}, \mathbf{I})$, and replace $\hat{s}_t^N$ in $\hat{\tau}_t^N$ with $s_t$ as condition on the current observation:

$$\hat{\tau}_t^N = \{(s_t, \hat{a}_t^N), (\hat{s}_{t+1}^N, \hat{a}_{t+1}^N), ..., (\hat{s}_{t+H}^N, \hat{a}_{t+H}^N)\}, \tag{4}$$

in which all the elements except $s_t$ are pure Gaussian noise. We further feed $\hat{\tau}_t^N$ into the backward process of diffusion to generate the subsequent trajectory:

$$p_\theta(\hat{\tau}_t^{i-1}|\hat{\tau}_t^i) = \mathcal{N}(\mu_\theta(\hat{\tau}_t^i, i) + \rho\nabla\mathcal{J}_\phi(\hat{\tau}_t^i, i), \beta_i\mathbf{I}), \tag{5}$$

$$\mu_\theta(\hat{\tau}_t^i, i) = \frac{\sqrt{\alpha^i}(1-\bar{\alpha}^{i-1})}{1-\bar{\alpha}^{i-1}}\hat{\tau}_t^i + \frac{\sqrt{\bar{\alpha}^{i-1}}\beta^i}{1-\bar{\alpha}^i}\hat{\tau}_t^{i,0}. \tag{6}$$

Here $\hat{\tau}_t^{i,0} = \psi_\theta(\hat{\tau}_t^i, i)$ represents the $\tau_t^0$ constructed from $\hat{\tau}_t^i$ at diffusion step $i$, $\psi_\theta(\cdot, \cdot)$ is a network for trajectory generation, $i \sim [1, N]$ is the diffusion step, $\rho$ represents the guidance scale, $\mathcal{J}_\phi(\cdot, \cdot)$ is a learned function to predict the return given any noisy trajectory $\hat{\tau}_t^i$ and the corresponding diffusion step $i$. We abbreviate $\hat{\tau}_t^0$ to $\hat{\tau}_t$ for convenience, $\hat{\tau}_t = \{(s_t, \hat{a}_t), (\hat{s}_{t+1}, \hat{a}_{t+1}), ..., (\hat{s}_{t+H}, \hat{a}_{t+H})\}$. $\hat{\tau}_t$ is considered as **the subsequent trajectory** following $s_t$. We take out the $\hat{a}_t$ in $\hat{\tau}_t$ as the action corresponding to the state $s_t$.

## 3.2 CONTRASTIVE MODULE

Although the Planning Module can independently generate the action responding to the environment, its performance is suppressed when the number of high-return trajectories is limited, as the Planning Module alone fails in making full use of the valuable insights implicated in low-return trajectories. As a solution, we propose a contrastive mechanism to improve the performance by constraining the states in a subsequent trajectory toward the high-return states and away from the low-return states. In the following parts, we first introduce the construction of contrastive sample sets (*i.e.*, sampling the positive and negative samples for contrasting), and then we explain how we perform the contrastive mechanism.

### 3.2.1 SAMPLE POSITIVE AND NEGATIVE STATES

The positive samples and negative samples are necessary before applying contrastive mechanism. For an arbitrary state $s_i \in \mathcal{S}$ in the offline dataset, we compute its return $v_i$ in advance, as is stated in Section 2.3. Then, we propose two strategies to sample positive sets and negative sets of an arbitrary state $s_t$:

**Sampling according to return (SR).** For an arbitrary state $s_t$ in the trajectory $\hat{\tau}_t$ generated by the Planning Module, we apply the theory of Thoma et al. (2020) to compute the possibility of an arbitrary state $s_i \in \mathcal{S}$ in the offline dataset is sampled as the positive sample and negative sample of state $s_t$:

$$p_{s_t}^+(v_i) = \frac{1}{1+e^{\sigma(\xi-v_i)}}, \tag{7}$$

$$p^-_{s_t}(v_i) = \frac{1}{1 + e^{\sigma(v_i - \zeta)}}, \tag{8}$$

where $v_i$ denotes the return of $s_i$. $p^+_{s_t}(v_i)$ and $p^-_{s_t}(v_i)$ denotes the probability of $s_i$ being grouped into positive sample and negative sample of $s_t$, correspondingly. $\sigma$, $\xi$ and $\zeta$ are the hyper-parameters extended from Thoma et al. (2020).

**Sampling according to return and dynamic consistency (SRD).** Though *sampling according to return* is easy to deploy, such a rough strategy neglects the transition probability between adjacent states in trajectories. Hence, for the sake of dynamic consistency, we additionally conduct MiniBatch K-Means clustering (Sculley, 2010) over states in the offline dataset. Given the cluster $\mathcal{C}_t$ to which $s_t$ belongs, we take the next state of all the states in $\mathcal{C}_t$ as the positive candidate set of $s_t$, $\mathcal{U}^+_t$. Note that we use the entire state space $\mathcal{S}$ as the negative candidate set of $s_t$, $\mathcal{U}^-_t$. Formally, we have $\mathcal{U}^+_t = \{s'|(s, a, s') \in \mathcal{D}, s \in \mathcal{C}_t\}, \mathcal{U}^-_t = \mathcal{S}$. Then, the positive and negative samples of $s_t$ can be sampled from $\mathcal{U}^+_t$ by Equation (7) and sampled from $\mathcal{U}^-_t$ by Equation (8), respectively.

### 3.2.2 CONSTRAIN THE TRAJECTORY WITH CONTRASTIVE LEARNING

To constrain the states in subsequent trajectories while avoiding the cost of running the whole backward denoising process, we leverage the noised trajectory in the diffusion backward process to reconstruct a neat trajectory, *i.e.*, $\hat{\tau}^{i,0}_t = \{(\hat{s}^{i,0}_t, \hat{a}^{i,0}_t), (\hat{s}^{i,0}_{t+1}, \hat{a}^{i,0}_{t+1}), ..., (\hat{s}^{i,0}_{t+H}, \hat{a}^{i,0}_{t+H})\}$ from $\tau^i_t$ for any arbitrary diffusion step $i$. Then, we extract states in $\hat{\tau}^{i,0}_t$ as $\mathcal{S}_{\hat{\tau}^{i,0}_t} = \{\hat{s}^{i,0}_{t+1}, \hat{s}^{i,0}_{t+2}, ..., \hat{s}^{i,0}_{t+H}\}$. For each state $\hat{s}^{i,0}_h \in \mathcal{S}_{\hat{\tau}^{i,o}_t}$, we sample $\kappa$ states as positive sample set $\mathcal{S}^+_h$ and $\kappa$ states as negative sample set $\mathcal{S}^-_h$ from the offline dataset.

Inspired by Schroff et al. (2015) and Sohn (2016), to apply contrastive learning to the scenario of multiple positive samples and impose aggressive constraints, we removed the positive sample term from the denominator polynomial in Equation (2) and propose the following equation to pull the states in the generated subsequent trajectory toward the high-return states and away from the low-return states:

$$\mathcal{L}^i_h = -\log \frac{\sum_{k=0}^{\kappa} \exp(\text{sim}(f(\hat{s}^{i,0}_h), f(s^+_h))/T)}{\sum_{k=0}^{\kappa} \exp(\text{sim}(f(\hat{s}^{i,0}_h), f(s^-_h))/T)}, \tag{9}$$

where $s^+_h \in \mathcal{S}^+_h$, $s^-_h \in \mathcal{S}^-_h$. $f(\cdot)$ represents the projection function, $T$ represents the temperature (Wang & Liu, 2021), and $\text{sim}(\cdot, \cdot)$ denotes the cosine similarity, which is computed as

$$\text{sim}(a, b) = \frac{a^\top b}{\|a\| \cdot \|b\|} . \tag{10}$$

It is worth noting that Equation (9) is differs from the standard InfoNCE loss (Oord et al., 2018). In practice, we found that removing positive samples from the denominator, as suggested by (Schroff et al., 2015) and (Sohn, 2016), allows for better utilization of negative samples (*i.e.*, low-return trajectories) and leads to better performance.

### 3.3 MODEL LEARNING

Recall that the action responding to state $s_t$ is one of the elements in the generated trajectory, which is influenced by the return predictor $\mathcal{J}_\phi(\cdot, \cdot)$ and constrained by contrastive learning. Therefore, we optimize our method from the perspective of trajectory generation, return prediction, and contrastive learning constrain.

Specifically, we optimize the trajectory generation by minimizing the Mean Square Error between the ground truth and neat trajectory predicted by $\psi_\theta(\cdot, \cdot)$ given any intermediate noisy trajectories as input:

$$\mathcal{L}_d = \mathbb{E}_{\tau_t \in \mathcal{D}, t > 0, i \sim [1,N]} \left[ \|\tau_t - \psi_\theta(\tau^i_t, i)\|^2 \right] , \tag{11}$$

where $i$ denotes the step of diffusion, $\tau^i_t$ is obtained in the $i$-th step of forward process.

We optimize the return predictor by minimizing the Mean Square Error between the predicted return $\mathcal{J}_\phi(\tau^i_t, i)$ and the ground-truth return $v_t$:

$$\mathcal{L}_v = \mathbb{E}_{\tau_t \in \mathcal{D}, t > 0, i \sim [1,N]} [\|v_t - \mathcal{J}_\phi(\tau^i_t, i)\|^2] . \tag{12}$$

We constrain the trajectory generation with a weighted contrastive loss:

$$\mathcal{L}_c = \mathbb{E}_{t>0, i\sim[1,N]} \left[ \sum_{h=t}^{t+H} \frac{1}{h+1} \mathcal{L}_h^i \right], \tag{13}$$

in which the coefficient $\frac{1}{h+1}$ decreases as $h$ increases since the importance gradually diminishes as it approaches the end of the planning horizon.

Hence, the overall objective function of ContraDiff can be written as a weighted sum of the aforementioned loss terms:

$$\mathcal{L} = \mathcal{L}_d + \mathcal{L}_v + \lambda_c \mathcal{L}_c, \tag{14}$$

where $\lambda_c$ is a hyper-parameter, which balances the importance of the contrastive module. Please note that the return predictor $\mathcal{J}_\phi(\cdot, \cdot)$ and $\psi_\theta(\cdot, \cdot)$ are independent, thus optimizing $\mathcal{J}_\phi(\cdot, \cdot)$ and $\psi_\theta(\cdot, \cdot)$ with $\mathcal{L}$ is identical to separately optimizing $\mathcal{J}_\phi(\cdot, \cdot)$ with $\mathcal{L}_v$ and $\psi_\theta(\cdot, \cdot)$ with $\mathcal{L}_d$ and $\mathcal{L}_c$. Please refer to the proof in Appendix A.8 for details.

The pseudo code of ContraDiff is presented in Appendix A.1, the details of hyper-parameters are available in Appendix A.11.

## 4 EXPERIMENTS

In this section, we evaluate the performance of ContraDiff across a wide variety of tasks. We first construct 27 sub-optimal datasets to evaluate the ability of ContraDiff and regular RL methods in cases where high-return samples are insufficient. Next, we demonstrate the advantages of ContraDiff by comparing ContraDiff with resampling-based methods. Further, we delve into more comprehensive experiments to analyze the key designs of ContraDiff. Note that results in tables of this section are reported over 50 random seeds. The best and the second-best results of each setting are marked as **bold** and underline, respectively.

### 4.1 EXPERIMENT SETTINGS

**Environments and datasets.** We evaluate ContraDiff on the locomotion tasks. Specifically, we evaluate the locomotion capability of ContraDiff on Halfcheetah, Hopper, Walker2d. For each environment, we train ContraDiff with various scales of offline datasets provided by D4RL (Fu et al., 2020), and test the performance of ContraDiff on the corresponding environments.

**Baselines.** We compare ContraDiff with the resampling-based methods, which focus on addressing the limited high-return trajectory issue. These methods are advantage-weighting (AW) (Hong et al., 2023a), return-weighting (RW) (Hong et al., 2023a), density-ratio weighting with advantage (AW-DW) (Hong et al., 2023b) and density-ratio weighting with uniform (U-DW) (Hong et al., 2023b). We also compare ContraDiff with other regular methods which focus on addressing the general offline reinforcement learning issues. These methods include newly proposed SOTA methods Decision Transformer (DT) (Chen et al., 2021), CDE (Cen et al., 2024), Trajectory Transformer (TT) (Janner et al., 2021), HD (Chen et al., 2024), Decision Stacks (Zhao & Grover, 2024), ReDiffuser (He et al., 2024), Diffuser (Janner et al., 2022), Decision Diffuser (DD) (Ajay et al., 2023).

**Implementation details.** We adopt U-Net (Ronneberger et al., 2015) as the denoise network $\psi_\theta(\cdot, \cdot)$ and the return predictor $\mathcal{J}_\phi(\cdot, \cdot)$, and adopt a linear layer with $Sigmoid$ as the activation function as the projector $f(\cdot)$. Our model is trained on a device with 4 NVIDIA A40 GPUs, Intel Gold 5220 CPU and 504G memory, optimized by Adam (Kingma & Ba, 2014) optimizer.

### 4.2 MAIN RESULTS

As we discussed previously, the proportion of high-return trajectories has a significant impact on the performance of the model. To further demonstrate the impact of the proportion of high-return samples on model performance and to evaluate the performance of ContraDiff under varying proportions of high-return samples, especially in scenarios with sparse high-return samples, we combine data within the dataset and create three types of datasets:

- **M-Exp**: mix the the trajectories in Medium and Expert.

- **MR-Exp**: mix the the trajectories in Med-Replay and Expert.

- **Rand-Exp** mix the trajectories in Expert and the trajectories sampled by random policy interacting with environment.

Table 1: Comparison of average normalized score between ContraDiff and regular offline RL methods on sub-optimal datasets, with $\pm$ denoting the standard deviation. The ratio denotes the ratio of trajectories from the Expert dataset.

| Environment | Dataset | Mix Ratio | CDE | HD | Decision Stacks | ReDiffuser | DT | TT | Diffuser | DD | ContraDiff-SR | ContraDiff-SRD |
|---|---|---|---|---|---|---|---|---|---|---|---|---|
| Halfcheetah | M-Exp | 0.1 | 42.4 ± 0.9 | 52.4 ± 3.2 | 11.4 ± 0.3 | 61.0 ± 3.3 | 58.6 ± 2.1 | 46.7 ± 1.1 | 71.5 ± 1.8 | 43.2 ± 0.9 | 72.1 ± 0.9 | **73.6 ± 1.2** |
| | | 0.2 | 41.9 ± 1.1 | 55.0 ± 1.5 | 25.7 ± 1.7 | 63.7 ± 0.9 | 46.8 ± 0.5 | 46.9 ± 1.8 | 80.3 ± 0.8 | 41.6 ± 2.2 | **81.3 ± 1.3** | 77.5 ± 0.9 |
| | | 0.3 | 44.0 ± 1.9 | 79.7 ± 1.2 | 35.0 ± 1.4 | 71.3 ± 2.7 | 70.9 ± 1.2 | 47.4 ± 1.1 | 81.7 ± 2.9 | 43.6 ± 3.3 | **82.8 ± 1.2** | 68.3 ± 1.1 |
| | MR-Exp | 0.1 | 32.5 ± 1.6 | 39.6 ± 0.8 | 29.1 ± 1.3 | **43.9 ± 0.6** | 7.5 ± 2.1 | 42.9 ± 2.5 | 38.1 ± 1.1 | 33.8 ± 1.6 | 37.8 ± 0.1 | 39.0 ± 0.5 |
| | | 0.2 | 34.1 ± 1.7 | 43.3 ± 2.9 | 36.2 ± 0.9 | 55.8 ± 3.1 | 6.7 ± 4.5 | 43.7 ± 1.7 | 46.2 ± 0.2 | 32.8 ± 0.6 | 50.6 ± 1.2 | **58.4 ± 0.9** |
| | | 0.3 | 35.2 ± 0.9 | 59.7 ± 2.1 | 41.0 ± 1.0 | 59.3 ± 1.6 | 6.1 ± 0.1 | 49.3 ± 2.2 | 57.1 ± 1.8 | 36.2 ± 1.1 | **60.3 ± 2.7** | 55.9 ± 1.5 |
| | Rand-Exp | 0.1 | 44.9 ± 4.0 | 1.9 ± 0.2 | 6.7 ± 0.9 | 42.8 ± 0.5 | 5.1 ± 0.0 | 7.7 ± 0.1 | 33.8 ± 1.8 | 13.8 ± 0.8 | 18.1 ± 1.1 | **48.0 ± 2.9** |
| | | 0.2 | 69.9 ± 1.6 | 5.5 ± 1.1 | 8.9 ± 0.2 | 50.1 ± 0.7 | 10.3 ± 4.2 | 16.8 ± 1.8 | **74.4 ± 1.5** | 8.5 ± 0.2 | 72.3 ± 0.7 | 65.0 ± 1.3 |
| | | 0.3 | 64.9 ± 4.2 | 5.9 ± 0.9 | 5.1 ± 0.1 | 59.9 ± 0.9 | 27.5 ± 6.8 | 5.9 ± 0.2 | 75.8 ± 2.3 | 13.9 ± 1.1 | 86.6 ± 1.8 | **88.7 ± 0.9** |
| Hopper | M-Exp | 0.1 | 52.8 ± 3.2 | 91.4 ± 2.1 | 93.2 ± 2.1 | 84.3 ± 1.4 | 27.0 ± 4.3 | 45.2 ± 1.2 | 82.3 ± 1.7 | 85.8 ± 0.7 | 87.1 ± 1.2 | **93.3 ± 1.1** |
| | | 0.2 | 57.5 ± 6.5 | 97.9 ± 4.2 | 73.1 ± 1.5 | 86.2 ± 2.5 | 24.9 ± 2.0 | 45.7 ± 0.8 | 89.4 ± 1.5 | 90.1 ± 0.1 | 97.9 ± 0.9 | **100.1 ± 0.9** |
| | | 0.3 | 60.1 ± 1.7 | 105.1 ± 0.3 | 43.9 ± 1.3 | 104.3 ± 2.7 | 20.6 ± 3.1 | 51.3 ± 1.3 | 104.8 ± 0.4 | 96.3 ± 0.6 | **106.0 ± 2.9** | 106.0 ± 1.4 |
| | MR-Exp | 0.1 | 51.3 ± 0.3 | 77.3 ± 2.2 | 77.5 ± 2.0 | 79.1 ± 2.3 | 48.2 ± 0.9 | 29.7 ± 0.9 | 63.2 ± 0.9 | 78.8 ± 1.2 | **80.3 ± 2.6** | 54.8 ± 1.9 |
| | | 0.2 | 72.3 ± 2.9 | **92.1 ± 1.5** | 71.1 ± 2.7 | 67.1 ± 1.2 | 46.6 ± 1.5 | 31.5 ± 4.3 | 69.5 ± 3.2 | 69.4 ± 4.3 | 73.9 ± 0.1 | 51.7 ± 0.8 |
| | | 0.3 | 86.1 ± 2.1 | 102.3 ± 3.2 | 99.4 ± 1.8 | 82.4 ± 0.9 | 55.2 ± 0.5 | 28.1 ± 1.9 | 69.7 ± 1.4 | **105.4 ± 3.2** | 65.4 ± 1.1 | 67.8 ± 1.9 |
| | Rand-Exp | 0.1 | 40.6 ± 3.2 | 3.2 ± 0.2 | 2.1 ± 0.9 | 40.2 ± 2.2 | 51.2 ± 1.9 | 2.1 ± 0.1 | 33.4 ± 1.4 | 0.9 ± 1.1 | **52.0 ± 0.7** | 46.9 ± 3.3 |
| | | 0.2 | 66.5 ± 5.2 | 17.2 ± 2.4 | 15.1 ± 0.3 | 49.2 ± 0.8 | 48.9 ± 2.1 | 2.0 ± 0.0 | 63.0 ± 0.4 | 1.1 ± 0.4 | 67.0 ± 2.2 | **75.3 ± 1.0** |
| | | 0.3 | 85.5 ± 1.1 | 28.9 ± 1.1 | 22.7 ± 2.1 | 36.1 ± 2.1 | 70.9 ± 1.5 | 2.1 ± 0.0 | 70.5 ± 1.2 | 1.8 ± 0.1 | **86.4 ± 1.5** | 83.3 ± 1.2 |
| Walker2d | M-Exp | 0.1 | 76.8 ± 1.6 | 88.1 ± 2.1 | 85.2 ± 0.4 | 105.8 ± 2.1 | 91.0 ± 1.0 | 82.1 ± 2.1 | 93.3 ± 0.6 | 49.9 ± 0.3 | 81.4 ± 0.4 | **107.1 ± 1.6** |
| | | 0.2 | 81.7 ± 2.6 | 101.6 ± 1.9 | 85.5 ± 0.6 | 99.5 ± 0.4 | 108.9 ± 0.2 | 82.0 ± 1.7 | 102.9 ± 1.7 | 56.2 ± 2.9 | **103.1 ± 1.2** | 82.0 ± 0.2 |
| | | 0.3 | 79.9 ± 1.6 | 101.9 ± 0.8 | 97.0 ± 1.2 | 96.8 ± 1.1 | 37.2 ± 0.5 | 81.5 ± 1.2 | 95.21 ± 0.3 | 31.7 ± 3.1 | **102.3 ± 1.5** | 97.0 ± 2.2 |
| | MR-Exp | 0.1 | 44.0 ± 2.3 | 79.8 ± 3.2 | 63.4 ± 2.0 | 84.9 ± 3.3 | 64.3 ± 1.9 | 45.2 ± 1.1 | 84.0 ± 1.0 | 66.8 ± 2.2 | **90.5 ± 2.2** | 61.4 ± 1.2 |
| | | 0.2 | 54.9 ± 1.9 | 88.4 ± 0.7 | 75.5 ± 0.4 | 89.7 ± 2.8 | 21.8 ± 4.4 | 41.1 ± 2.0 | 83.3 ± 0.7 | 84.6 ± 1.4 | **90.8 ± 0.6** | 75.9 ± 0.7 |
| | | 0.3 | 43.5 ± 4.5 | 87.2 ± 1.1 | 80.0 ± 1.1 | 80.6 ± 3.2 | 37.2 ± 2.5 | 17.1 ± 1.2 | 86.9 ± 2.2 | 70.6 ± 1.2 | **93.2 ± 0.3** | 80.4 ± 1.1 |
| | Rand-Exp | 0.1 | 14.0 ± 2.3 | 0.2 ± 0.0 | 10.2 ± 0.1 | 3.2 ± 0.2 | 10.5 ± 0.1 | 5.1 ± 0.1 | 14.6 ± 0.8 | 0 | **20.2 ± 1.3** | 14.7 ± 1.9 |
| | | 0.2 | 54.9 ± 1.9 | 23.4 ± 0.4 | 3.2 ± 0.3 | 49.7 ± 3.1 | **89.1 ± 0.9** | 5.6 ± 0.7 | 48.8 ± 3.2 | 55.2 ± 1.9 | 57.4 ± 0.7 | 51.0 ± 0.9 |
| | | 0.3 | 73.5 ± 2.1 | 42.2 ± 0.1 | 15.3 ± 0.1 | 77.4 ± 1.1 | **85.3 ± 3.2** | 3.9 ± 5.4 | 48.9 ± 2.9 | 60.1 ± 3.1 | 78.4 ± 1.2 | 48.3 ± 2.2 |

We select the ratio of trajectories from the Expert dataset from {0.1, 0.2, 0.3}, resulting in 27 mixed sub-optimal datasets in total. Returns of sub-optimal Halfcheetah datasets are illustrated in Figure 7. From the dataset level, for Medium and Med-Replay, mixing with Expert data effectively increases the proportion of high-return samples. Additionally, since Medium and Med-Replay are both collected based on trained policy, whereas Random is collected based on a randomly initialized policy (Wu et al., 2021), after mixing with Expert data, Rand-Exp exhibits an overall lower proportion of high-return samples compared to M-Exp and MR-Exp. Furthermore, within the datasets, a higher proportion of mixed Expert data corresponds to a higher proportion of high-return samples. Using these datasets, we compared ContraDiff with resampling-based methods and regular methods.

### 4.2.1 COMPARISON WITH REGULAR METHODS

Results of ContraDiff and baselines on the datasets are illustrated in Table 1, in which ContraDiff-SR denotes the states used for contrasting are sampled with SR strategy and ContraDiff-SRD denotes the states used for contrasting are sampled with SRD strategy. From Table 1, we can observe that our method ContraDiff achieves the optimal and sub-optimal results on 25 out of 27 datasets, showing the advantage of ContraDiff in situations with sub-optimal datasets. More specifically, (1) In most cases, the performance of offline RL methods declines as the ratio decreases, where a smaller ratio indicates fewer high-return trajectories. Moreover, compared with the performance in M-Exp, most baselines have a performance decline when trained with Rand-Exp, which has fewer high-return trajectories than Med-Expert. These results validate that the performance of offline RL methods is influenced by the proportion of high-return trajectories in the dataset; (2) ContraDiff demonstrates more significant improvement in Rand-Exp than M-Exp and MR-Exp, for instance, it outperforms the best baseline by 2.2 on Hopper-M-Exp-0.2, but it outperforms the best baseline by 8.8 on Hopper-Rand-Exp-0.2. As a brief recap, ContraDiff makes better use of low-quality samples than the baseline method, and demonstrates significant superiority in the case with limited number of high-return trajectories.

### 4.2.2 COMPARISON WITH RESAMPLING METHODS

Some previous studies proposed to address the issue of limited high-return trajectories have proven effective. To further evaluate the effectiveness of ContraDiff, we compare it with these methods in more challenging scenarios where the agent is trapped in low-return states. Our main objective here is to investigate the ability of ContraDiff to handle such situations. For fairness in comparison, we use Diffuser, which is also the backbone of ContraDiff, as the backbone for all methods. Specifically, we combined Diffuser with AW (Hong et al., 2023a), RW (Hong et al., 2023a), AW-DW (Hong et al., 2023b) and U-DW (Hong et al., 2023b) as our baselines.

Comparison results are summarized in Table 2. As can be observed, all resampling method aids in enhancing the effectiveness of Diffuser. Even though, the performance of ContraDiff exceeds all of the resampling methods in most cases, demonstrating its effectiveness under sub-optimal conditions. Moreover, similar to Table 1, as the proportion of high-return samples increases, ContraDiff shows a more significant performance improvement compared to the baseline methods, indicating that utilizing low-return samples can effectively help the model escape from low-return areas.

Table 2: Comparison of average normalized score between ContraDiff and resampling-based baseline methods on various sub-optimal datasets, with ± denoting the standard deviation. Ratio denotes the the ratio of trajectories from the Expert dataset.

| Environment | Dataset | Mix Ratio | Diffuser | AW+Diffuser | RW+Diffuser | AW-DW+Diffuser | U-DW+Diffuser | ContraDiff-SR | ContraDiff-SRD |
|---|---|---|---|---|---|---|---|---|---|
| Halfcheetah | M-Exp | 0.1 | 53.2 ± 2.1 | 63.8 ± 0.4 | 66.6 ± 1.7 | 25.9 ± 2.9 | 35.8 ± 2.1 | 63.2 ± 2.9 | **69.6 ± 3.1** |
| | | 0.2 | 66.3 ± 0.4 | 77.5 ± 2.2 | 77.1 ± 1.3 | 39.7 ± 1.9 | 32.9 ± 0.7 | **80.3 ± 1.3** | 67.5 ± 0.9 |
| | | 0.3 | 72.9 ± 1.9 | 80.3 ± 1.2 | **82.3 ± 1.3** | 49.4 ± 0.6 | 44.2 ± 3.2 | 72.8 ± 1.2 | 62.3 ± 1.1 |
| | MR-Exp | 0.1 | 30.1 ± 1.1 | 40.1 ± 3.2 | 38.2 ± 1.1 | 27.2 ± 2.7 | 4.9 ± 1.9 | **40.8 ± 1.3** | 33.0 ± 2.9 |
| | | 0.2 | 39.9 ± 3.2 | 41.1 ± 1.7 | 47.6 ± 1.7 | 25.9 ± 1.6 | 15.4 ± 2.3 | 40.3 ± 1.2 | **48.4 ± 1.9** |
| | | 0.3 | 44.6 ± 2.8 | 52.7 ± 1.1 | **59.9 ± 1.7** | 46.3 ± 2.2 | 20.1 ± 2.1 | 51.9 ± 1.6 | 50.2 ± 0.9 |
| | Rand-Exp | 0.1 | 23.8 ± 1.6 | **43.4 ± 0.9** | 32.2 ± 1.1 | 2.9 ± 0.7 | 3.8 ± 0.9 | 19.5 ± 1.7 | 35.8 ± 1.9 |
| | | 0.2 | 55.7 ± 0.5 | 60.6 ± 2.3 | 40.1 ± 3.9 | 23.7 ± 0.2 | 11.3 ± 0.8 | 55.3 ± 0.7 | **61.5 ± 1.7** |
| | | 0.3 | 66.3 ± 1.3 | 53.4 ± 1.1 | 78.2 ± 2.1 | 37.4 ± 2.1 | 24.1 ± 3.9 | 67.7 ± 1.2 | **79.1 ± 0.5** |
| Hopper | M-Exp | 0.1 | 79.1 ± 2.7 | 83.9 ± 2.1 | 85.5 ± 2.2 | 48.9 ± 1.3 | 39.2 ± 1.1 | 77.1 ± 1.5 | **91.3 ± 2.4** |
| | | 0.2 | 92.9 ± 0.7 | 80.1 ± 1.7 | 94.4 ± 0.2 | 37.8 ± 0.1 | 35.1 ± 1.0 | 103.9 ± 3.7 | **98.5 ± 2.8** |
| | | 0.3 | 97.2 ± 2.4 | 103.9 ± 2.2 | 100.8 ± 1.4 | 47.9 ± 0.2 | 45.7 ± 2.9 | **104.9 ± 1.8** | 102.6 ± 3.2 |
| | MR-Exp | 0.1 | 61.4 ± 1.3 | 23.1 ± 1.0 | 63.3 ± 2.6 | 13.2 ± 2.1 | 1.5 ± 0.4 | **77.3 ± 2.4** | 58.8 ± 2.9 |
| | | 0.2 | 59.1 ± 1.2 | 25.7 ± 2.1 | 69.1 ± 0.3 | 31.9 ± 0.5 | 1.6 ± 0.1 | **70.3 ± 1.8** | 42.7 ± 2.8 |
| | | 0.3 | 62.5 ± 2.1 | 60.3 ± 3.1 | **72.0 ± 2.8** | 55.9 ± 0.7 | 7.9 ± 1.3 | 61.4 ± 1.3 | 65.8 ± 2.3 |
| | Rand-Exp | 0.1 | 23.4 ± 1.4 | 12.3 ± 1.0 | 31.2 ± 2.8 | 10.6 ± 3.3 | 0.7 ± 0.6 | **45.1 ± 2.7** | 34.0 ± 1.7 |
| | | 0.2 | 45.2 ± 2.1 | 14.4 ± 0.3 | 41.9 ± 0.3 | 24.4 ± 0.2 | 1.9 ± 0.9 | 47.0 ± 2.2 | **50.2 ± 1.1** |
| | | 0.3 | 49.1 ± 1.7 | 12.3 ± 0.9 | 69.2 ± 2.1 | 32.3 ± 1.9 | 5.9 ± 0.7 | **76.9 ± 0.9** | 56.9 ± 2.2 |
| Walker2d | M-Exp | 0.1 | 79.1 ± 2.1 | 72.1 ± 2.8 | 69.7 ± 0.14 | 7.4 ± 0.4 | 9.4 ± 0.9 | 74.4 ± 1.4 | **101.1 ± 2.1** |
| | | 0.2 | 85.3 ± 0.7 | 90.3 ± 3.2 | 83.0 ± 2.3 | 21.3 ± 0.6 | 14.7 ± 1.5 | **92.8 ± 0.7** | 69.2 ± 1.2 |
| | | 0.3 | 88.7 ± 1.9 | 95.2 ± 1.9 | 87.6 ± 0.7 | 69.9 ± 0.4 | 10.0 ± 1.3 | **99.3 ± 1.5** | 91.0 ± 2.2 |
| | MR-Exp | 0.1 | 68.3 ± 1.3 | 59.5 ± 1.2 | 66.7 ± 2.5 | 2.7 ± 1.1 | 2.9 ± 0.6 | **74.5 ± 2.7** | 65.1 ± 1.2 |
| | | 0.2 | 74.2 ± 1.9 | 80.3 ± 3.2 | 75.9 ± 2.5 | 11.4 ± 3.2 | 1.1 ± 0.2 | **82.8 ± 3.2** | 65.9 ± 1.6 |
| | | 0.3 | 80.2 ± 3.2 | 89.4 ± 0.9 | 83.4 ± 3.2 | 13.9 ± 0.7 | 0.8 ± 0.3 | **91.3 ± 0.3** | 71.4 ± 1.7 |
| | Rand-Exp | 0.1 | 4.3 ± 0.3 | 11.6 ± 3.3 | 10.4 ± 4.1 | 4.9 ± 1.3 | 1.9 ± 0.2 | 12.2 ± 2.1 | **32.2 ± 1.6** |
| | | 0.2 | 19.3 ± 2.0 | 32.9 ± 1.9 | 29.6 ± 2.9 | 7.7 ± 0.8 | 2.3 ± 3.1 | **55.4 ± 1.7** | 30.4 ± 2.2 |
| | | 0.3 | 36.1 ± 1.9 | 53.9 ± 1.7 | 50.1 ± 0.7 | 35.5 ± 2.2 | 2.1 ± 2.2 | **61.6 ± 0.9** | 45.3 ± 1.2 |

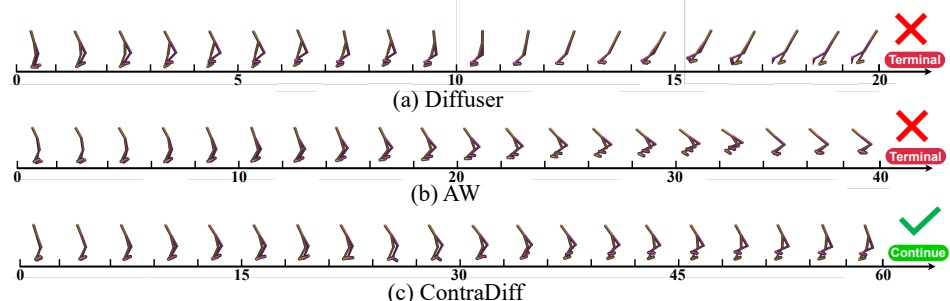

(a) Diffuser

(b) AW

(c) ContraDiff

Figure 3: How different models escape from low-return regions on Walker2d-Rand-Exp-0.3.

Additionally, we visualized how Diffuser, AW and ContraDiff escape from low-return regions on Walker2d-Rand-Exp-0.3. As shown in Figure 3, given a similar initial state which is on the verge of falling backward, only ContraDiff successfully recovers from this posture and achieves long-term, healthy interactions (lasting over 60 steps), indicating that the contrastive mechanism helps ContraDiff in escaping from low-return states. Even though AW can interact with the environment for a longer duration compared to Diffuser, it ultimately fails to achieve the more prolonged interactions similar to ContraDiff. Recalling that AW only focuses on the high-return trajectories via reweighing, the comparison between AW and ContraDiff more intuitively highlights the importance of information in low-return samples.

## 4.3 FURTHER INVESTIGATION

To further investigate the performance of ContraDiff, we conduct ablation study, and analyze the state-reward distribution as well as the reward distribution.

### 4.3.1 ABLATION STUDIES

We have the following variants to conduct ablation study:

- **ContraDiff-N**: only apply the samples with high-return to train the model.

- **ContraDiff-C**: remove contrastive mechanism from ContraDiff, *i.e.*, remove $\mathcal{L}_c$ from Equation (14).

The results are summarized in Figure 4, from which we can conclude the following key findings:

(1) **Applying only the high-return samples in training diminishes benefits in some cases.** Compared with ContraDiff, ContraDiff-N is trained solely with high-return samples. Surprisingly, it achieves a lower performance than ContraDiff in all 9 tasks, indicating that low-return trajectories offer beneficial information for model training.

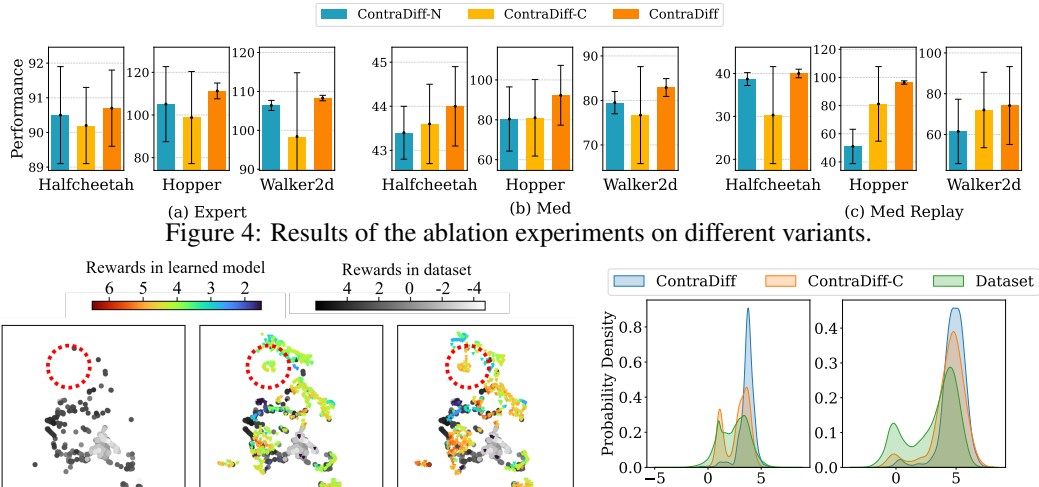

Figure 4: Results of the ablation experiments on different variants.

Figure 5: The distribution of state and reward. It is better to view in color mode. ContraDiff achieves higher rewards in out-of-distribution areas (circled with red).

Figure 6: Distribution of rewards on (a) Walker2d-Med-Replay and (b) Halfcheetah-Med-Replay.

(2) **The contrastive mechanism benefits the performance.** ContraDiff-C, which eliminates the contrastive mechanism from ContraDiff, exhibits poorer performance across all nine tasks than ContraDiff. This suggests that the contrastive mechanism indeed provides benefits.

### 4.3.2 STATE-REWARD DISTRIBUTION ANALYSIS

To illustrate the advantages of ContraDiff more intuitively, we randomly collect the (state, reward) pairs from the offline dataset of Walker2d-Med-Replay and the (state, reward) pairs collected when ContraDiff and ContraDiff-C interact with the environment. The results are shown in Figure 5, in which each scatter represents a state mapped by UMAP (McInnes et al., 2018), and its color denotes the reward gained in the corresponding state. From the results illustrated in Figure 5, we can observe that: (1) there are more red and yellow dots in Figure 5(c) than (b). That indicates that the model achieves better rewards by utilizing low-return samples more thoroughly with contrasting mechanisms; (2) In out-of-distribution states (circled with red), ContraDiff gains higher rewards than ContraDiff-C, which indicates that contrastive mechanism has potential in tackling out-of-distribution issue.

### 4.3.3 REWARD DISTRIBUTION ANALYSIS

The contrast mechanism in ContraDiff plays a significant role in more efficiently utilizing low-return samples. We believe that a deeper reason lies in the impact of the contrast mechanism in the learning of the diffusion model. According to Section 3.3, three crucial components of ContraDiff are the trajectory generation of the diffusion model, return prediction, and the contrast mechanism. To validate our assumption, we remove the return prediction of ContraDiff and ContraDiff-C (*i.e.*, remove $\mathcal{J}_\phi$ from Equation (5)), leverage them to generate subsequent trajectory. Then we apply the actions in the generated trajectory to interact with the environment Walker2d-Med-Replay and HalfCheetah-Med-Replay, and visualize the distribution of reward during the interaction, as shown in Figure 6. It can be observed that the ContraDiff has a higher probability density on high rewards in both cases. That indicates the effect of the contrast mechanism essentially increasing the proportion of high-return trajectories generated by the diffusion model. As demonstrated in Equation (5), with the support of both generated high-return trajectories and the return predictor, ContraDiff achieves sound performance.

We have also conducted experiments to investigate the computational requirements of our method (Appendix A.9), impact of the size of contrastive set (Appendix A.10), hyper-parameters analysis (Appendix A.11), ablation study about the weighted contrastive loss (Appendix A.12) and the analysis of averaged completed tasks (Appendix A.13), please refer to the Appendix for details.

## 5 RELATED WORKS

### 5.1 DIFFUSION FOR DECISION-MAKING

With diffusion models demonstrate their powerful generalization capabilities across different fields, many works have introduced diffusion models into reinforcement learning for decision-making. Some methods use diffusion models for generating actions directly (Ada et al., 2023; Wang et al., 2022; Chen et al., 2022a; Chi et al., 2023). For instance, Diffusion Q-learning (Wang et al., 2022) designs the policy as a diffusion model and improves it with double Q-learning architecture. Following this, SRDPs (Ada et al., 2023) adopts diffusion policies for out-of-distribution problems. Some other

methods predict the future trajectories with diffusion models for decision-making. Diffuser (Janner et al., 2022) models trajectories as sequences of state-action pairs. Based on this, Decision Diffuser (Ajay et al., 2023) proposes to predict state sequences with a diffusion model conditioned on historical information, and adopts a reverse dynamic model to predict actions. Though these methods have gain significant achievements, they neglect the differences between high-return and low-return samples, and fails in making full use of low-return samples.

## 5.2 Offline Reinforcement Learning with Imbalanced Datasets

In many cases, offline RL datasets tend to have a limited number of high-return samples, while low-return samples are much more abundant, which is generally considered as the data imbalance issue (Choi et al., 2024; Yan et al., 2024). Many works have been proposed to tackle this problem, often by sample reweighing. Some works reweight samples at the trajectory level. For instance, Hong et al. (2023a) suggests that sampling can be reweighted to prioritize high-return trajectories, and proposes return-weighted (RW) and advantage-weighted (AW) as solutions. Based on this, Hong et al. (2023b) proposes Density-ratio Weighting (DW), which designs a learnable weight for reinforcement learning objectives. Some other methods reweight samples at a more fine-grained level, $i.e.$, at action or state level. LAPO (Chen et al., 2022b) maps actions into a latent space, which is further optimized through advantage-weighting; and more recently, A2PR (Liu et al., 2024) obtains high-advantage actions from an augmented behavior policy. Differently, SAW (Lyu et al., 2022) reweights the future expected states and uses them as the goal for the policy. Although they focus on high-return samples through reweighing, they overlook the valuable information contained in low-return samples. In contrast, our method ContraDiff, leveraging the contrastive learning mechanism, fully exploits the information embedded in low-return samples.

## 5.3 Contrastive Learning in Reinforcement Learning

Most methods adopt contrastive learning (CL) to learn better representations, such as state representations (Laskin et al., 2020; Qiu et al., 2022) and task representations (Yuan & Lu, 2022; Agarwal et al., 2020). For instance, Laskin et al. (2020) proposes to learn image representations via CL, Qiu et al. (2022) propose to learn the transition with CL, Yuan & Lu (2022) and Agarwal et al. (2020) applies CL to enhance the task representations to distinguish between different tasks. Some works apply CL in other ways. For instance, Laskin et al. (2022) utilize CL to learn behavior representations to encourage behavioral diversity, while QGPO (Lu et al., 2023) adopts CL for better score function of Diffusion-based RL methods. In contrast, ContraDiff adopts CL for better usage of low-return samples.

## 6 Conclusion and Discussion

In this paper, we introduce ContraDiff for offline RL, which combines contrastive mechanism to make better uses low-return trajectories and addresses the challenge of limited high-return samples. Different from the previous works which apply contrastive learning to enhance the representation, we perform contrastive learning over the return of states. Specifically, we apply diffusion to generate the subsequent trajectory for planning, and constrain the states in the generated trajectory toward the states with high returns and away from the states with low returns to improve the base distribution. In that way, the actions taken by the agent are always toward the high-return states, which makes the agent gain better performance in the online evaluation. We evaluate ContraDiff on 27 sub-optimal datasets, where the results demonstrate that our ContraDiff is able to make better uses low-return samples and achieves outstanding performance. The ablation studies and investigations further substantiated the rationality of ContraDiff.

Although ContraDiff has achieved outstanding performance, there are still many areas for improvement and scalability. For instance, our current contrastive learning is applied to states, thus the ContraDiff framework can only be extended to trajectory generation methods for now, such as action or state-action pairs, or even at the latent of trajectories. We leave applying the contrastive mechanism to other other methods in the future work.

## 7 Acknowledgments

This work was supported by the National Natural Science Foundation of China (Grants No. 62307020 and 62322603), the National Key R&D Program of China (2022ZD0114804), and the Shanghai Municipal Science and Technology Major Project (2021SHZDZX0102).

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

## A    APPENDIX

### A.1    PSEUDO CODE OF CONTRADIFF.

---
**Algorithm 1** Training
---
1: **while** not converged **do**
2:    $\boldsymbol{\tau}_t, v_t \sim \mathcal{D}$.
3:    $i \sim [1, N]$.
4:    Generate $\boldsymbol{\tau}_t^i$.
5:    Reconstruct $\boldsymbol{\tau}_t$ as $\hat{\boldsymbol{\tau}}_t^{i,0} = \psi_\theta(\boldsymbol{\tau}_t^i, i)$.
6:    Calculate loss $\mathcal{L}_d$ with Equation (11).
7:    Calculate loss $\mathcal{L}_v$ with Equation (12).
8:    Extract states in $\hat{\boldsymbol{\tau}}_t^{i,0}$ as $\mathcal{S}_{\hat{\boldsymbol{\tau}}_t^{i,0}} = \{\hat{\boldsymbol{s}}_{t+1}^{i,0}, \hat{\boldsymbol{s}}_{t+2}^{i,0}, ..., \hat{\boldsymbol{s}}_{t+H}^{i,0}\}$.
9:    **for** $\hat{\boldsymbol{s}}_h^{i,0}$ in $\mathcal{S}_{\hat{\boldsymbol{\tau}}_t^{i,0}}$ **do**
10:       Sample $\mathcal{S}^+$ and $\mathcal{S}^-$ according to Section 3.2.1.
11:       Calculate $\mathcal{L}_h^i$ using Equation (8).
12:    **end for**
13:    Calculate $\mathcal{L}_c$ using Equation (13).
14:    Calculate $\mathcal{L}$ using Equation (14).
15:    Update model by taking gradient decent with $\mathcal{L}$.
16: **end while**
---

---
**Algorithm 2** Planning
---
**Require:** ContraDiff $\psi_\theta(\cdot, \cdot)$, return-to-go predictor $\mathcal{J}_\phi(\cdot, \cdot)$, guidance scale $\rho$, co-variances $\boldsymbol{\Sigma}^i$.
1: $t \leftarrow 1$.
2: **while** not done **do**
3:    Observe state $\boldsymbol{s}_t$; sample $\boldsymbol{\tau}_t^N \sim \mathcal{N}(\boldsymbol{0}, \boldsymbol{I})$
4:    **for** $i = N, N-1, ..., 1$ **do**
5:       Predict return-to-go with $\mathcal{J}_\phi(\hat{\boldsymbol{\tau}}_t^i, i)$.
6:       Sample $\hat{\boldsymbol{\tau}}_t^{i-1}$ using Equation (5).
7:    **end for**
8:    Extract $\hat{\boldsymbol{a}}_t$ form $\hat{\boldsymbol{\tau}}^0$.
9:    Interact with environment using action $\hat{\boldsymbol{a}}_t$.
10:    $t \leftarrow t + 1$.
11: **end while**
---

### A.2    ILLUSION OF EQUATION (7) AND EQUATION (8).

Example of Equation (7) and Equation (8) are visualized in Figure 8. As can be observed in Figure 8, our modified influence functions are designed to leave a blank in the middle area deliberately, which is different from (Thoma et al., 2020). The underlying reason is that not all states are supposed to be contrastive samples. Nevertheless, our modified influence functions collapse into modified influence functions in (Thoma et al., 2020) if we set $\xi = \zeta$.

### A.3    RESULTS ON REGULAR DATASETS

We have also compared ContraDiff with baseline methods on regular datasets. The results of ContraDiff and baseline methods are summarized in Table 3 and Table 4.

From Table 3 and Table 4, we can observe that: (1) Compared with all the baseline methods, ContraDiff achieves the best or the second-best performance on 5 out of 9 locomotion tasks (HalfCheetah, Hopper, and Walker2d) and achieves the best or the second-best performance on all the two high-dimensional manipulation tasks (Kitchen), demonstrating the outstanding performance of ContraDiff under periodic settings. Moreover, ContraDiff achieves the best performance on 2 out of 3 navigation tasks, demonstrating the excellent ability of ContraDiff in long-term planning. (2) Compared to the methods with similar backbones, Diffuser outperforms Diffuser in all 14 tasks, and

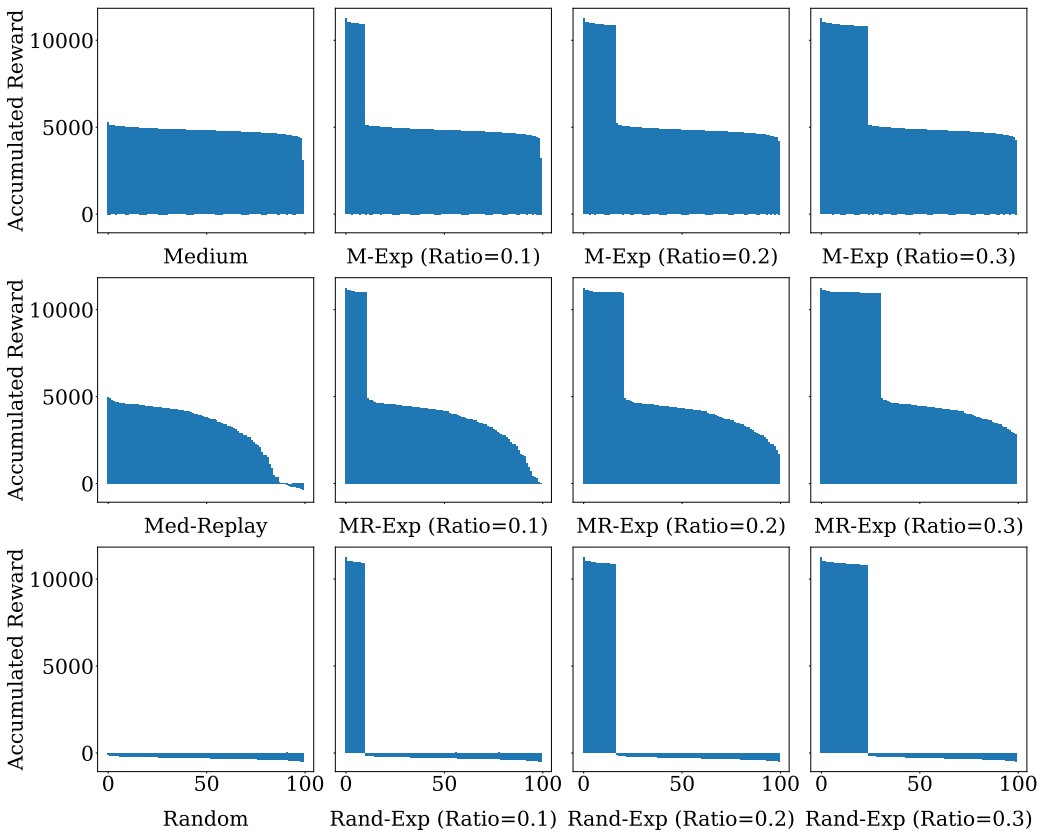

Figure 7: Returns distribution of Medium, Med-Replay and Random datasets of Halfcheetah mixed with different ratios of Expert data.

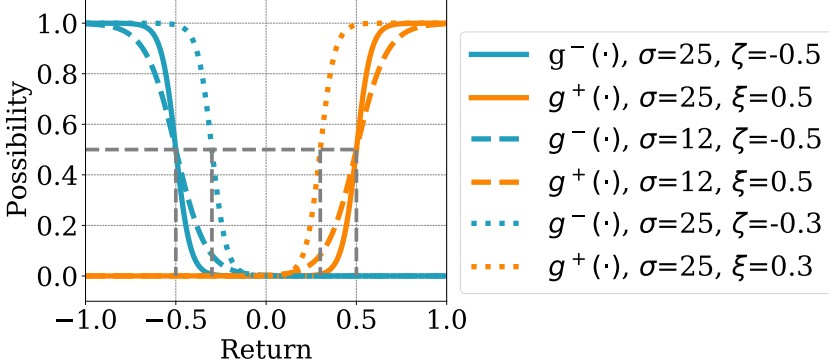

Figure 8: Illusion of Equation (7) and Equation (8)

Table 3: Comparison of average normalized score on Mujoco environments between ContraDiff and baselines, with ± denoting the standard deviation.

| Dataset | Environment | CQL | IQL | DT | TT | MOPO | AW-IQL | RW-IQL | AW+DW-IQL | U+DW-IQL | AW-BC | RW-BC | CDE | Decision Stack | Diffuser | DD | ContraDiff-SR | ContraDiff-SRD |
|---|---|---|---|---|---|---|---|---|---|---|---|---|---|---|---|---|---|---|
| Med-Expert | Halfcheetah | 91.6 | 86.7 | 86.8 | 95.0 | 63.3 | 94 | 93 | 93.9 | 93.7 | 92 | 92 | 75.6 ± 7.2 | 95.7 ± 0.3 | 88.9 ± 0.3 | 90.6 ± 1.3 | 92.0 ± 0.4 | 89.9 ± 0.6 |
| Med-Expert | Hopper | 105.4 | 91.5 | 107.6 | 110.0 | 23.7 | 99 | 101 | 110.8 | 81.0 | 110 | 110 | 108.6 ± 4.8 | 107.0 ± 3.2 | 103.3 ± 1.3 | 111.8 ± 1.8 | 112.4 ± 1.2 | 106.4 ± 1.3 |
| Med-Expert | Walker2d | 108.8 | 109.6 | 108.1 | 101.9 | 44.6 | 110 | 109 | 109.7 | 109.7 | 108 | 108 | 107.7 ± 10.4 | 108.0 ± 0.1 | 106.9 ± 0.2 | 108.8 ± 1.7 | 108.2 ± 0.4 | 108.7 ± 1.4 |
| Medium | Halfcheetah | 44.0 | 47.4 | 42.6 | 46.9 | 42.3 | 47 | 47 | 47.9 | 47.7 | 42 | 42 | 43.3 ± 2.9 | 47.8 ± 0.4 | 42.8 ± 0.3 | 49.1 ± 1.0 | 43.9 ± 0.9 | 42.9 ± 0.2 |
| Medium | Hopper | 58.5 | 66.3 | 67.6 | 61.1 | 28.0 | 57 | 57 | 61.7 | 62.5 | 56 | 54 | 51.2 ± 3.7 | 76.7 ± 4.2 | 74.3 ± 1.4 | 79.3 ± 3.6 | 92.3 ± 2.6 | 88.6 ± 0.5 |
| Medium | Walker2d | 72.5 | 78.3 | 74.0 | 79.0 | 17.8 | 69 | 66 | 75.8 | 80.8 | 70 | 71 | 73.8 ± 4.8 | 83.6 ± 0.3 | 79.6 ± 0.5 | 82.5 ± 1.4 | 82.9 ± 0.5 | 82.2 ± 1.1 |
| Med-Replay | Halfcheetah | 45.5 | 44.2 | 36.6 | 41.9 | 53.1 | 44 | 44 | 44.6 | 44.6 | 38 | 40 | 32.7 ± 0.8 | 41.1 ± 0.1 | 37.7 ± 0.5 | 39.3 ± 4.1 | 40.0 ± 1.1 | 36.6 ± 2.9 |
| Med-Replay | Hopper | 95.0 | 94.7 | 82.7 | 91.5 | 67.5 | 84 | 86 | 99.9 | 79.7 | 72 | 67 | 73.9 ± 2.4 | 89.5 ± 4.2 | 93.6 ± 0.4 | 100.0 ± 0.7 | 96.4 ± 1.1 | 95.5 ± 0.9 |
| Med-Replay | Walker2d | 77.2 | 73.9 | 66.6 | 82.6 | 39.0 | 47 | 37 | 62.6 | 65.1 | 56 | 56 | 77.1 ± 3.1 | 80.7 ± 1.5 | 70.6 ± 1.6 | 75.0 ± 4.3 | 84.2 ± 1.2 | 75.3 ± 1.7 |

Table 4: Comparison of average normalized score on Maze2d and Kitchen between ContraDiff and baselines, with ± denoting the standard deviation.

| Dataset | Environment | CQL | IQL | DT | TT | MOPO | Decision Stack | Diffuser | DD | ContraDiff-SR | ContraDiff-SRD |
|---|---|---|---|---|---|---|---|---|---|---|---|
| U-Maze | Maze2d | 5.7 | 47.4 | 9.2 | 25.4 | 13.6 | 111.3 ± 12.2 | 113.9 ± 3.1 | 0.0 | 142.9 ± 2.2 | 122.1 ± 1.4 |
| Medium | Maze2d | 5.0 | 34.9 | 9.6 | 23.3 | 33.3 | 111.7 ± 2.4 | 121.5 ± 2.7 | 0.0 | 140 ± 0.7 | 115.7 ± 0.7 |
| Large | Maze2d | 12.5 | 58.6 | 10.4 | 27.7 | 0.0 | 171.6 ± 13.4 | 123.0 ± 6.4 | 0.0 | 131.5 ± 3.2 | 129.9 ± 1.2 |
| Mixed | Kitchen | 52.4 | 51.0 | 20.9 | 31.1 | 0.0 | - | 42.5 ± 1.9 | 65.0 ± 2.8 | 65.0 ± 1.3 | 32.5 ± 2.2 |
| Partial | Kitchen | 51.2 | 46.3 | 35.2 | 32.9 | 0.0 | - | 40.0 ± 3.1 | 57.0 ± 2.5 | 58.0 ± 1.9 | 40.0 ± 3.1 |

outperforms DD in 11 tasks, which demonstrates the benefits of introducing the constrastive mechanism. (3) We can observe that ContraDiff exhibits more improvement in Medium and Med-Replay datasets than the Med-Expert datasets. Since Med-Expert datasets have more high-return samples, they offer abundant information for methods like Diffuser to learn, thus they can achieve better results. However, both Med and Med-Replay have more low-return samples than Med-Expert, which increases the difficulty of learning a good policy. (4) Compared to the sub-optimal datasets in Table 1, the improvement of ContraDiff-SRD in the regular scenarios is not as pronounced as that of ContraDiff-SR. Even though, ContraDiff-SRD outperforms Diffuser in 11 out of 14 situations. We believe this is because, in the regular dataset, it is easier for the model to learn the transitions between different states, making the guarantee of dynamic consistency (*i.e.*, ContraDiff-SRD) relatively less important in this context. In other words, we recommend using ContraDiff-SR in regular scenarios, and use ContraDiff-SRD in challenging scenarios. For more detailed discussions, please refer to Appendix A.4.

## A.4    WHICH STRATEGY TO SELECT FOR THE CONSTRUCTION OF CONTRASTIVE SAMPLE?

We provide two implementations to construct contrastiev samples, namely ContraDiff-SR and ContraDiff-SRD. As described in Section 3.2.1, ContraDiff-SR pulls the generated trajectories towards the globally high-return states, while ContraDiff-SRD pulls the generated trajectories towards the transition-able high-return states. This may result in a slight decrease in comparative effectiveness, but it ensures dynamic consistency among contrastive samples.

Although the results indicate that each method has its advantages, there are also patterns to follow when making a choice. We first visualize the samples and then determine which implementation to choose based on the visualization results. For example, let's consider Halfcheetah-Rand-Exp-0.1 and Walker2d-Rand-Exp-0.3. The states with their returns are visualized in Figure 9, where crosses represent the expert data. From this visualization, we can obtain some prior information about dynamic consistency.

As shown in Figure 9 (a), most of the high-return states are far away from the low-return states. From this observation, we can infer that starting from any one of the original states from the Halfcheetah-Random dataset, it is difficult to transition to the expert data, *i.e.*, the high-return states marked with a cross. Under this situation, pulling the generated trajectories with ContraDiff-SR will introduce uncertainty, as the corresponding state transitions are unachievable. Therefore, we adopt ContraDiff-SRD for Halfcheetah-Rand-Exp-0.1.

In contrast, as shown in Figure 9 (b), the mixed expert data of Walker2d-Rand-Exp-0.3 shares a similar distribution pattern as the original states from the Walker2d-Random dataset. In this situation, we adopt ContraDiff-SR for more direct constrain.

## A.5    COMPATIBILITY STUDY

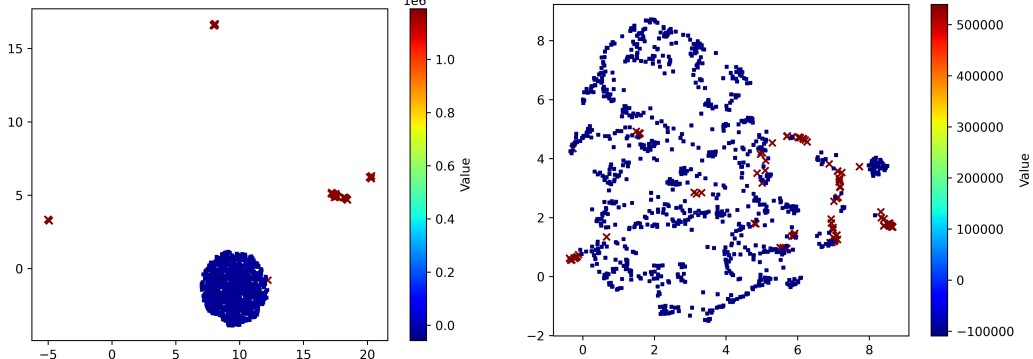

Figure 9: States distribution of (a) Halfcheetah-Rand-Exp-0.1 and (b) Walker2d-Rand-Exp-0.3, with cross representing the expert data.

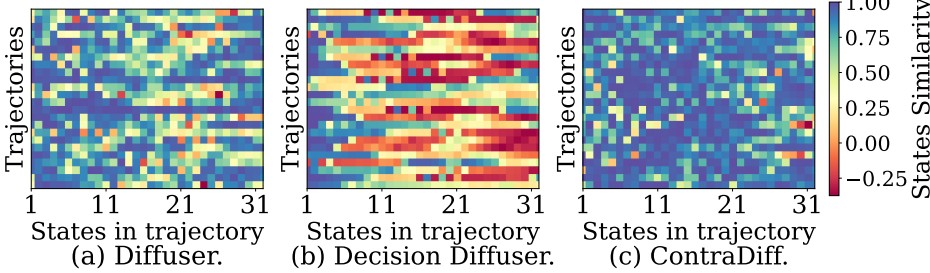

Figure 10: The similarities between the states in the generated trajectories and actual states. The generated states of ContraDiff are more similar with the actual states, demonstrating the better long-term dynamic consistency.

As we discussed in Section 3.1, our ContraDiff is build based on Diffuser. To validate the compatibility of the contrastive mechanism of ContraDiff, we transplant it to Decision Diffuser (DD) (Ajay et al., 2023), and evaluate and compare the improvement on three environments. The improvements are summarized in Table 5, in which $DD^+$ and $Diffuser^+$ denote the improvement of introducing contrast mechanism in DD and Diffuser correspondingly. As we can observe, $DD^+$ achieves noticeable improvement in 2 out of 3 tasks and $Diffuser^+$

Table 5: The improvements of the normalized score after transplanting the contrastive mechanism of ContraDiff to Decision Diffuser ($DD^+$) and Diffuser ($Diffuser^+$), with $\pm$ denoting the variance.

| Dataset | Environment | $DD^+$ | $Diffuser^+$ |
|---|---|---|---|
| Med-Expert | Halfcheetah | 0 | 3.1±0.4 |
| Med-Expert | Hopper | 5.4±1.2 | 9.1±1.2 |
| Med-Expert | Walker2d | 6.5±0.9 | 1.3±0.4 |

gains improvement in 3 out of 3 tasks, which demonstrates the portability of the contrast mechanism of ContraDiff. Interestingly, $DD^+$ is unable to achieve any improvement in Halfcheetah-Med-Expert. This could be attributed to the separated training (sampling) of states and actions in DD, which results in a failure to effectively model their joint distribution.

## A.6 PLAN-EXECUTION CONSISTENCY ANALYSIS.

We use the plan-execution consistency to denote the similarity between the states in the planned trajectory and the states encountered by the agent during its interaction with the environment. It reflects the models' capability in modeling the environment. To investigate the plan-execution consistency of ContraDiff, we randomly take 24 trajectories generated by Diffuser, Decision Diffuser, and ContraDiff. For each generated trajectory, we take the states of consecutive 32 steps and compute the similarity between each generated state and the actual state of the same step provided by the environment. Thus, there are $24 \times 32$ similarity returns for each model, which corresponds to a similarity matrix as the subgraphs in Figure 10 illustrated. Each line in the subgraphs of Figure 10 represents a generated trajectory, and the grids of each line represent the similarity of the states in

the generated trajectory and the states provided by the environment. From Figure 10, we can observe that: (1) Most grids in Figure 10 (c) are blue, which denotes that most generated states are consistent with the actual states; (2) Figure 10 (c) contains more blue grids than Figure 10 (a) and (b), which denotes that ContraDiff has better plan-execution consistency than Diffuser and Decision Diffuser. Since the difference between ContraDiff and Diffuser is the contrastive module, combining Figure 5 and Figure 10, we can conclude that the contrative module benefits the plan-execution consistency of ContraDiff and makes ContraDiff gain high rewards in both in-distribution and out-of-distribution situations.

### A.7 VISUALIZATION OF POSITIVE AND NEGATIVE SAMPLES.

We randomly sample a subset of positive samples (states with high returns) and negative samples (states with low returns), as is shown in Figure 11. It can be observed that an agent in a state corresponding to a high return tends to be in a position more conducive to walking or running, such as standing upright; correspondingly, an agent with a state corresponding to a low return will be in a position that is hard to walk, such as having already fallen down or about to fall down. This is reasonable, since poses such as standing upright are more conducive to walking or running, which causes the agent to continue moving and results in a higher return, while poses such as having fallen or about to fall cause the environment to give a stop signal, which results in a lower return.

### A.8 OPTIMIZING $\mathcal{J}_\phi(\cdot, \cdot)$ WITH EQUATION (14)

Suppose we have the diffuison model $\psi_\theta(\cdot)$ parameterized by $\theta$, and the return predictor $\mathcal{J}_\phi$ parameterized by $\phi$. Following Equation (14), we have

$$\mathcal{L} = \mathcal{L}_d + \mathcal{L}_v + \lambda_c \mathcal{L}_c. \tag{15}$$

Further,

$$\mathcal{L}_d = \mathbb{E}_{\boldsymbol{\tau}_t \in \mathcal{D}, t>0, i \sim [1,N]} \left[ \| \boldsymbol{\tau}_t - \psi_\theta(\boldsymbol{\tau}_t^i, i) \|^2 \right], \tag{16}$$

$$\mathcal{L}_v = \mathbb{E}_{\boldsymbol{\tau}_t \in \mathcal{D}, t>0, i \sim [1,N]} [\| \mathcal{J}_\phi(\boldsymbol{\tau}_t^i, i) - v_t \|^2]. \tag{17}$$

The training process can be viewed as a procedure of calculating gradients of all the parameters and updating them, specifically,

$$\nabla\theta = \frac{\partial \mathcal{L}}{\partial \theta} \tag{18}$$

$$= \frac{\partial \mathcal{L}_d}{\partial \theta} + \frac{\partial \mathcal{L}_v}{\partial \theta} + \lambda_c \frac{\partial \mathcal{L}_c}{\partial \theta} \tag{19}$$

$$= \frac{\partial \mathcal{L}_d}{\partial \theta} + \lambda_c \frac{\partial \mathcal{L}_c}{\partial \theta}, \tag{20}$$

$$\nabla\phi = \frac{\partial \mathcal{L}}{\partial \phi} \tag{21}$$

$$= \frac{\partial \mathcal{L}_d}{\partial \phi} + \frac{\partial \mathcal{L}_v}{\partial \phi} + \lambda_c \frac{\partial \mathcal{L}_c}{\partial \phi} \tag{22}$$

$$= \frac{\partial \mathcal{L}_v}{\partial \phi}. \tag{23}$$

Thus, calculating the gradients of $\theta$ with $\mathcal{L}$ is equal to calculate $\theta$ with $\mathcal{L}_d$ and $\mathcal{L}_c$, calculating the gradients of $\phi$ with $\mathcal{L}$ is equal to calculate $\phi$ with $\mathcal{L}_v$, *i.e.*, optimizing the return predictor $\mathcal{J}_\phi(\cdot, \cdot)$ with Equation (14) is equal to optimizing it with Equation (12) only.

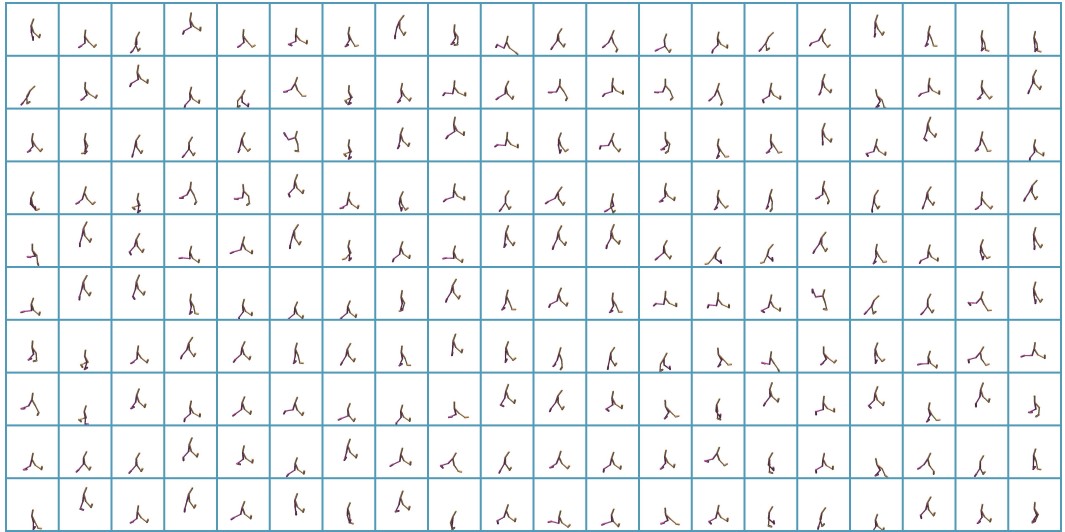

(a) Agents with high-return states.

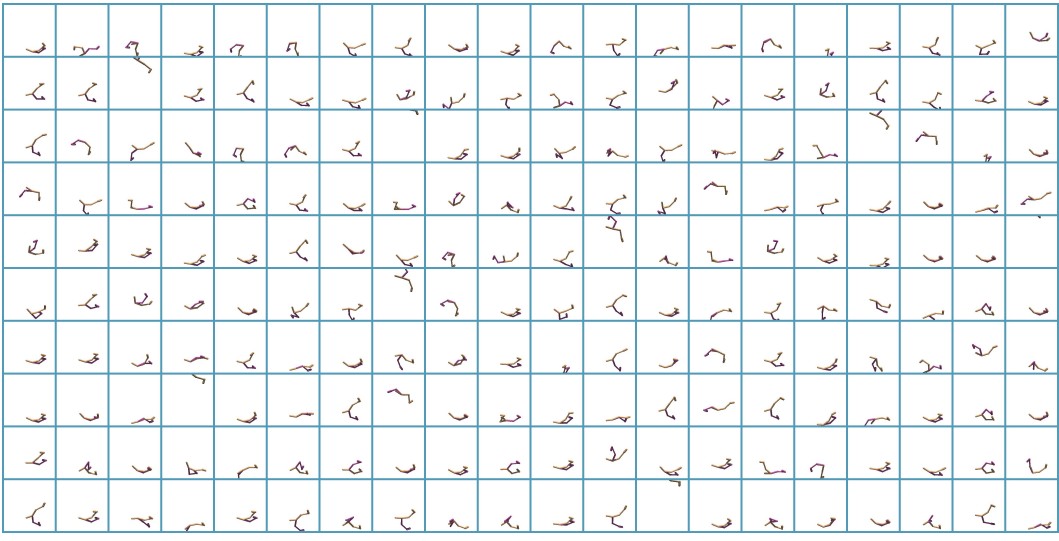

(b) Agents with low-return states.

Figure 11: Visualization of positive samples (states with high returns) and negative samples (states with low returns) in Walker2d-Med-Replay.

Table 6: Comparison of computational requirements.

| - | Decision Diffuser | Diffuser | ContraDiff-SR | ContraDiff-SRD |
|---|---|---|---|---|
| CPU (%) | 107.6 | 106.4 | 107.7 | 112.4 |
| Memory (GB) | 6.9 | 6.9 | 7.0 | 13.9 |
| Training time (Seconds every 1000 iterations) | 92 | 98 | 112 | 122 |

## A.9 COMPUTATIONAL REQUIREMENTS

We have quantified the impact of sample selection and the incorporation of contrastive learning on computational requirements in our method, as shown in Table 6.

It can observed that even with the introduction of sampling and contrastive learning, ContraDiff-SR requires a comparable amount of resources and training time to other methods with similar backbone. For ContraDiff-SRD, sampling based on dynamic consistency requires more resources. This is partly due to the dynamic consistency considerations, resulting in the requirement for more

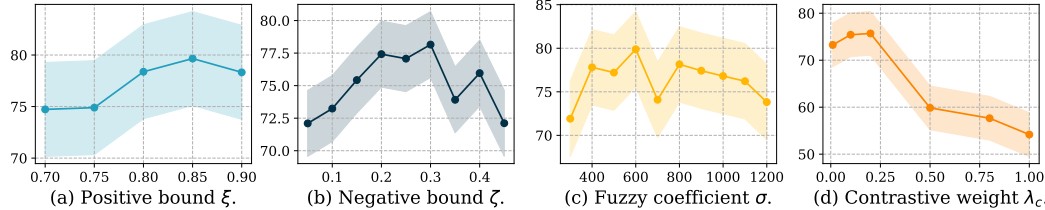

(a) Positive bound $\xi$.  (b) Negative bound $\zeta$.  (c) Fuzzy coefficient $\sigma$.  (d) Contrastive weight $\lambda_c$.

Figure 12: The impact of hyper-parameters.

CPU cores (5 more than other methods). Additionally, the introduction of cache matrices consumes more memory (7GB, but actually we can cache them in CPU and conduct the sampling in CPU if it is necessary).

Despite that, the resources and training time are still manageable. Moreover, additional resources are only required during the training phase. During the inference phase, our resource requirements are consistent with other methods (*e.g.*, Diffuser).

## A.10 IMPACT OF THE SIZE OF CONTRASTIVE SET

Table 7: Impact of the size of contrastive set.

| Mix Ratio | 8 | 16 | 32 | 64 | 128 |
|---|---|---|---|---|---|
| 0.1 | 35.4 ± 2.2 | 41.5 ± 2.6 | 48.0 ± 2.9 | 46.9 ± 1.1 | 48.9 ± 0.9 |
| 0.2 | 39.1 ± 2.9 | 69.6 ± 2.1 | 77.6 ± 0.9 | 73.9 ± 0.9 | 78.5 ± 0.6 |
| 0.3 | 73.9 ± 1.7 | 84.8 ± 1.1 | 88.7 ± 0.9 | 90.4 ± 1.7 | 88.3 ± 1.3 |

Generally, we set the size of contrastive set to 32. We illustrated the impact of the size of contrastive set on our method using Halfheetah-Rand-Exp in Table 7. We can observe, the performance of ContraDiff gradually increases when increasing the size within 32. However, as the size of the contrastive samples increases further, the performance gradually stabilizes. Therefore, the size of the contrastive set is easy to determine.

## A.11 HYPER-PARAMETERS.

ContraDiff imports the following 4 additional hyper-parameters: positive bound ($\xi$), negative bound ($\zeta$), fuzzy coefficient ($\sigma$), loss weight of contrastive learning ($\lambda_c$). For locomotion tasks, we select $\xi \in \{0.55, 0.65, 0.85\}$, select $\zeta \in \{0.05, 0.1, 0.2, 0.35, 0.4\}$, select $\lambda_c \in \{0.1, 0.01, 0.001\}$. $\sigma$ is select between $[4, 16] \times 10^2$. For Maze2d tasks, we select $\xi \in \{0.1, 0.6, 5\}$, select $\zeta \in \{0.01, 0.02, 0.2\}$. We set $\sigma$ as $1 \times 10^8$ and set $\lambda_c$ as 0.1 for Maze2d tasks. We use the other hyper-parameters as in the paper of Diffuser (Janner et al., 2022).

Note that even there are 4 new hyper-parameters, searching for the optimal hyper-parameters is easy. On the one hand, these hyper-parameters are not interdependent, so we can search for the optimal value of each hyper-parameter separately. On the other hand, analysis of the hyper-parameters indicates that their impact on model performance is moderate, as discussed below.

We have visualized the impact of hyper-parameters in Figure 12. As is summarized in Figure 12, (a) The choice of hyper-parameters has a minimal impact on performance, with variations remaining within 7%. (b) The performance changes smoothly around the peak region as illustrated in Figure 12, indicating tuning the hyper-parameters towards the increasing performance will obtain optimal performance. Therefore, tuning hyper-parameters of CDifffuser is relatively easy, costing little time. (c) The performance of the model does indeed change with variations of hyper-parameters, as the selection of positive and negative samples and the weight of the contrastive loss influence the performance. However, as we discussed in (a), the hyper-parameters are relatively easy to tune. Therefore, in most cases, hyper-parameters do not become a bottleneck limiting model performance. Still, fur-

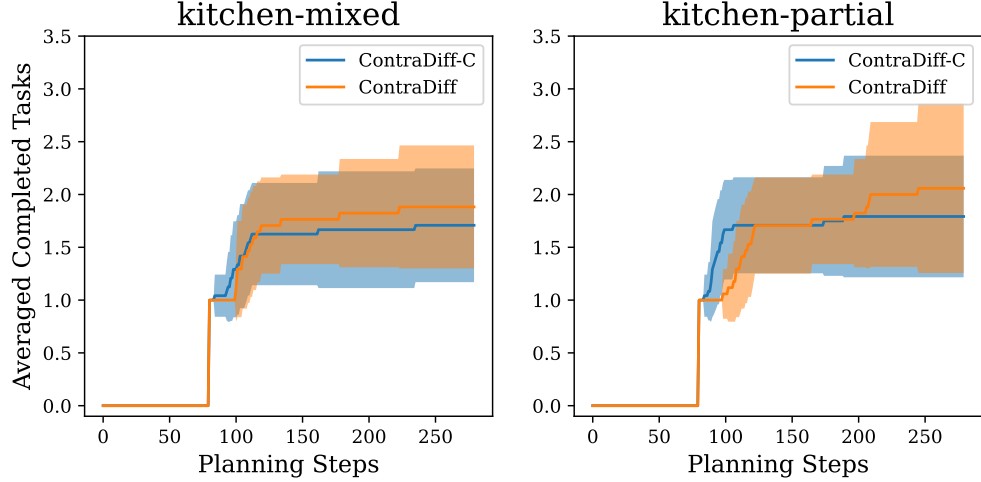

Figure 13: The averaged completed tasks of ContraDiff-C and ContraDiff.

ther optimizing the design of the hyper-parameters is one of the tasks we will focus on in our future research.

## A.12 ABLATION STUDY ABOUT EQUATION (13)

The weighted contrastive loss Equation (13) is designed to emphasize the initial part of the generation, as we use the initial action in generated trajectories to interact with the environment. Additionally, we have conducted ablation studies on the weighted contrastive loss, where ContraDiff-mean represents ContraDiff with an average-weighted contrastive loss. As is shown in Table 8, the weighted contrastive loss brings better performance in all of the situations.

Table 8: Performance comparison of ContraDiff-mean and ContraDiff. ContraDiff-mean represents ContraDiff with an average-weighted contrastive loss.

| Mix Ratio | ContraDiff-mean | ContraDiff |
|-----------|-----------------|------------|
| 0.1 | 34.2 ± 1.2 | **48.0 ± 2.9** |
| 0.2 | 71.1 ± 1.9 | **72.3 ± 0.7** |
| 0.3 | 82.9 ± 0.3 | **88.7 ± 0.9** |

## A.13 ANALYSIS OF AVERAGED COMPLETED TASKS

We additionally compared the number of completed tasks of ContraDiff and ContraDiff-C on Kitchen-Mixed and Kitchen-Partial to further demonstrate the advantages of ContraDiff. As is visualized in Figure 13, as the planning steps increase, ContraDiff shows more completed tasks than ContraDiff-C, which intuitively demonstrates the advantages of the contrastive mechanism in sparse reward environments.

## A.14 ANALYSIS OF RL LOSS AND CONTRASTIVE LEARNING LOSS

Recall the the diffusion loss (RL loss, Equation (11)), the contrastive loss Equation (13) in Section 3. Note that the diffusion loss (RL loss, Equation (11)) aims at modeling the dataset distribution. However, the closer the distance to high-return states and the farther the distance to low-return states, the smaller the value of Equation (13).

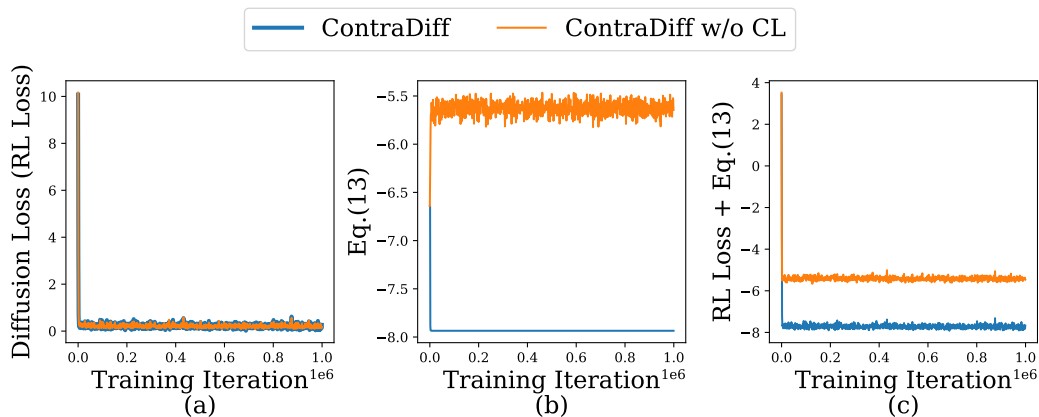

Figure 14: The (a) Diffusion loss (*i.e.*, RL loss, Equation (11)), (b) value of Equation (13), and (c) RL loss + value of Equation (13) on Walker2d-Rand-Exp-0.1.

As is shown in Figure 14(a), regardless of the use of contrastive learning, the RL loss shows a downward trend and eventually stabilizes. However, for the value of Equation (13) shown in Figure 14(b), when contrastive learning is not used, the contrastive loss remains large and fluctuates significantly. On the other hand, when contrastive learning is applied, the contrastive loss decreases. Moreover, ContraDiff shows a lower value on RL Loss + Equation (13) than ContrasDiff w/o CL in Figure 14(c). This demonstrates that (1) the RL loss and contrastive loss are independent, with the contrastive loss serving as a meaningful regularization term; and (2) contrastive learning indeed maximizes the distance from low-return trajectories while minimizing the distance to high-return trajectories, as further corroborated by the analyses in Section 4.3.

