# OpenReview forum: "ContraDiff: Planning Towards High Return States via Contrastive Learning"
_ICLR.cc/2025/Conference — ICLR 2025 Poster_

### Official Review · Reviewer_Ku5d · 2024-11-03

**Soundness:** 3
**Presentation:** 3
**Contribution:** 3
**Rating:** 6
**Confidence:** 3

**Summary:**

This paper introduces a novel approach, ContraDiff, designed to enhance offline reinforcement learning (RL) by leveraging low-return trajectories in a contrastive learning framework. Offline RL's performance often depends on the presence of high-return trajectories, which are frequently sparse in datasets. ContraDiff addresses this challenge by classifying trajectories into high- and low-return groups, treating them as positive and negative samples, respectively. The approach applies a contrastive mechanism that pulls the agent's trajectories toward high-return states while pushing them away from low-return ones. This strategy enables low-return trajectories to serve as a guiding force to avoid suboptimal regions, enhancing policy performance.

**Strengths:**

The paper presents a compelling approach to enhancing offline reinforcement learning (RL) through a novel contrastive framework, which incorporates low-return trajectories in ways that previous approaches overlook. ContraDiff’s use of a contrastive learning framework to distinguish between high- and low-return trajectories is an innovative application of contrastive learning within the offline RL domain. Instead of merely reweighting high-return samples, the method leverages low-return trajectories as guiding forces, creating a “counteracting force” that helps the policy avoid suboptimal states. This approach represents a shift from traditional methods that focus primarily on high-return trajectories, expanding the value of underutilized low-return data.

**Weaknesses:**

* While the paper presents contrastive learning as a means to exploit low-return data, it could better clarify why this mechanism is theoretically optimal for avoiding low-return states compared to other potential approaches, such as weighting adjustments or imitation-based filtering.

* The authors are not clear enough in describing the use of weighted contrast loss to constrain trajectory generation

**Questions:**

* In the 3.3 model learning section, I noticed that the optimized trajectory generation by minimizing the Mean Square Error between the ground truth and neat trajectory predicted. Therefore, the diffusion should denoise the data from the noisy data completely. I think it's more expensive to train like this. Can you show some experiments comparing the cost of training?
* ContraDiff planning towards high return states, leading policy improvements. It is very similar to some offline RL methods, such as SAW[1], A2PR[2], LAPO[3]. Can you add some discussion with these methods in the related works or more experiments comparison?
* Did you experiment with setting different thresholds for what qualifies as “low” or “high” return? If so, how did varying these thresholds impact the learned policy or the model’s ability to generalize to novel tasks?

References：
[1] Lyu, Jiafei, et al. "State advantage weighting for offline RL." arXiv preprint arXiv:2210.04251 (2022).

[2] Liu, Tenglong, et al. "Adaptive Advantage-Guided Policy Regularization for Offline Reinforcement Learning." In International Conference on Machine Learning (ICML). PMLR, 2024.

[3] Chen, Xi, et al. "Latent-variable advantage-weighted policy optimization for offline rl." arXiv preprint arXiv:2203.08949 (2022).

---

> ### Author Response · Authors · 2024-11-21
> **Responses to Reviewer Ku5d (Q1-Q4)**
>
> **Q1.** While the paper presents contrastive learning as a means to exploit low-return data, it could better clarify why this mechanism is theoretically optimal for avoiding low-return states compared to other potential approaches, such as weighting adjustments or imitation-based filtering. \
> **R1.** Thank you for your comment. RL methods based on weighting adjustments or imitation-based filtering usually assign weights to states or trajectories according to their returns, aiming to prioritize high-return samples during the optimization process and ignore low-return samples. In other words, these methods do not actively attempt to distance the model from low-return samples. In contrast, our approach uses contrastive learning, treating low-return samples as negative examples and high-return samples as positive examples. This not only brings generated trajectories closer to high-return trajectories but also actively pushes them away from low-return samples. The comparison with the reweighting methods (summarized in Table 2)  demonstrates the effectiveness of our proposed approach. We also designed ablation studies in Section 4.3.1 to demonstrate the effectiveness of utilizing low-return states.
>
> **Q2.** The authors are not clear enough in describing the use of weighted contrast loss to constrain trajectory generation. \
> **R2.**   Thank you for your comment. The weighted contrast loss is formulated in Eq.(13), which is further used for contractive learning, as formulated in Eq.(14). Details of  Eq.(13) and  Eq.(14) are discussed in Section 3.3. Specifically, at time step $ t_1 $, the agent obtains the observation from the environment and generates a future trajectory of $ H $ steps, $[ (s_h, a_h) ]_{t \leq h \leq h+H}$. Considering that the importance of predictions decreases as the time step extends further into the future, we apply a weighting scheme to the contrastive learning loss for the next $ H $ steps. For the contrastive learning loss at time step $ h $, we assign a weight of $ 1/(h+1) $.
>
> **Q3.** In the 3.3 model learning section, I noticed that the optimized trajectory generation by minimizing the Mean Square Error between the ground truth and neat trajectory predicted. Therefore, the diffusion should denoise the data from the noisy data completely. I think it's more expensive to train like this. Can you show some experiments comparing the cost of training? \
> **R3.** Thank you for your comment. The cost of training with the Mean Square Error between the ground truth and neat trajectory predicted (i.e., Eq.(11)) is the same as predicting the noise. Following EDP [1] and the official repo of Diffuser [2], we adopt the one-step denoising method to directly predict the original trajectory within one step rather than denoising the data from the noisy data completely, for additional supervisory information during the training phase. However, during the testing phase, we still generate trajectories (i.e., future plans) through the complete reverse denoising process.
>
> [1] Kang, B., Ma, X., Du, C., Pang, T., & Yan, S. (2024). Efficient diffusion policies for offline reinforcement learning. Advances in Neural Information Processing Systems, 36. \
> [2] Janner, M., Du, Y., Tenenbaum, J. B., & Levine, S. (2022). Planning with diffusion for flexible behavior synthesis. arXiv preprint arXiv:2205.09991.
>
> **Q4.** ContraDiff planning towards high return states, leading policy improvements. It is very similar to some offline RL methods, such as SAW[1], A2PR[2], LAPO[3]. Can you add some discussion with these methods in the related works or more experiments comparison?
> **R4.** Thank you for your suggestion! We have added discussions of SAW[1], A2PR[2], and LAPO[3] to enrich the Related Works section.
> We ported SAW to Diffuser (Diffuser-SAW) for experimentation. Since A2PR is not applicable to Diffuser, we conducted experiments with A2PR on our suboptimal datasets. We were unable to perform experiments related to LAPO because the code for LAPO was not provided in the paper. The comparison of ContraDiff with Diffuser-SAW and LAPO on Walker2d-Rand-Exp is shown in the table below. It can be observed that ContraDiff outperforms Diffuser-SAW and A2PR in all cases, demonstrating the advantage of ContraDiff.
>
>
> | Mix Ratio | Diffuser-SAW | A2PR  | ContraDiff |
> |-----------|--------------|-------|---------------------|
> | 0.1       | 18.6 ± 1.8   | 13.6 ± 4.3 | **20.2 ± 1.3**          |
> | 0.2       | 52.9 ± 3.3   | 21.8 ± 0.9 | **57.4 ± 0.7**          |
> | 0.3       | 61.3 ± 1.6   | 22.9 ± 1.4 | **78.4 ± 1.2**          |

---

> ### Author Response · Authors · 2024-11-21
> **Responses to Reviewer Ku5d (Q5)**
>
> **Q5.** Did you experiment with setting different thresholds for what qualifies as “low” or “high” return? If so, how did varying these thresholds impact the learned policy or the model’s ability to generalize to novel tasks? \
> **R5.** Thank you for your question. \
> (1) We have conducted the experiments of thresholds for positive and negative samples (i.e., the thresholds for what qualifies as “low” or “high” return) and hyperparameters (ξ, ζ, σ)  in Appendix A.11.  In summary,  analysis of the hyper-parameters indicates that their impact on model performance is moderate, and searching for the optimal hyper-parameters is easy. Please refer to Appendix A.11 for more detailed discussions.
> (2) The choice of threshold on novel tasks can be made a priori based on the state return in the dataset. For example, one can visualize the state-return distribution of the dataset, and take the return with the most dramatic return change as the threshold. For example, thresholds can be identified as turning points by calculating the rate of change (e.g., differences) or the second-order rate of change (acceleration)[1,2]. After that, we can fix the threshold and adjust $\sigma$ for better results. \
>
>
> [1] Satopaa, V., Albrecht, J., Irwin, D., & Raghavan, B. (2011, June). Finding a" kneedle" in a haystack: Detecting knee points in system behavior. In 2011 31st international conference on distributed computing systems workshops (pp. 166-171). IEEE. \
> [2] Silverman, B. W. (2018). Density estimation for statistics and data analysis. Routledge.

---

> ### Author Response · Authors · 2024-11-25
> **Responses to Reviewer Ku5d**
>
> Dear Reviewer Ku5d,
>
> We have provided detailed responses to your concerns many days ago, we hope these responses have adequately addressed your concerns. As the discussion phase is coming to an end, we sincerely request your further responses.
>
> If we have resolved your issues, please consider raising your score to the positive side. If you have any further questions, please feel free to share them with us! Any valuable feedback is crucial for improving our work.
>
> Best regards, \
> The Authors

---

> > ### Comment · Reviewer_Ku5d · 2024-11-26
> > **Official Comment by Reviewer Ku5d**
> >
> > Thanks for the effort of the authors. Most of my concerns have been addressed, but I have one more question. I see that the experiments in the paper all need to include expert data, and I'm curious how the performance of ContraDiff is without the expert dataset.

---

> > > ### Author Response · Authors · 2024-11-27
> > > **Responses to Reviewer Ku5d**
> > >
> > > Dear reviewer, \
> > > We are glad that we have addressed most of your concerns. Here are our responses:
> > >
> > >
> > > **Q1.** The performance of ContraDiff without the expert dataset. \
> > > **R1.** Thank you for your positive feedback. We introduced a small portion of expert data into each dataset to evaluate ContraDiff's performance in handling suboptimal datasets, in which high-return samples are limited. We provided the performance of ContraDiff without additional expert samples in the Mujoco, Maze2d, and Kitchen datasets in Table 3 and Table 4 from Appendix A.3. It can be observed that without expert samples, ContraDiff achieved the best or second-best results in 4 out of 6 Mujoco tasks and in all Maze2d and Kitchen tasks. This more comprehensively demonstrates the advantages of ContraDiff and indicates that it also performs well in sparse reward scenarios. Please refer to Table 3 and Table 4 in Appendix A.3 in our paper for details.

---

> ### Author Response · Authors · 2024-11-28
> **Request for Further Discussion**
>
> Dear Reviewer Ku5d,
>
> We have provided detailed responses to your concerns days ago, and have provided the performance of ContraDiff without the expert dataset accordingly. We hope our responses have adequately addressed your additional concern. As the discussion phase is coming to an end, we sincerely request your further responses.
>
> If we have resolved your issue, please consider raising your score to the positive side. If you have any further questions, please feel free to share with us! We would appreciate your further discussion on our paper.
>
> Best regards,
> The Authors

---

> ### Author Response · Authors · 2024-12-01
> **Sincere Requests for Further Discussions**
>
> Dear Reviewer Ku5d,
>
> Thank you for your effort in reviewing our paper! We are glad that we have addressed most of you concerns. We have provided the performance of ContraDiff without expert data accordingly several days ago, and we hope our responses have adequately addressed your additional concern. As the discussion phase is coming to an end, we sincerely request your further responses.
>
> If we have resolved your issue, please consider raising your score to the positive side. If you have any further questions, please feel free to share with us! We would appreciate your further discussion on our paper.
>
> Best regards,  \
> The Authors

---

> > ### Comment · Reviewer_Ku5d · 2024-12-02
> > **Official Comment by Reviewer Ku5d**
> >
> > Thanks for your additional experiments and effort. My concerns have been addressed. I have updated my score.

---

> > > ### Author Response · Authors · 2024-12-02
> > > **Response to Reviewer Ku5d**
> > >
> > > Dear Reviewer Ku5d,
> > >
> > >
> > > Thank you for your effort in reviewing our paper, and thank you for acknowledging our work! Your valuable comments and suggestions have significantly contributed to improving our paper.
> > >
> > >
> > > Best regards, \
> > > The Authors

---

### Official Review · Reviewer_DdY7 · 2024-11-03

**Soundness:** 2
**Presentation:** 2
**Contribution:** 2
**Rating:** 6
**Confidence:** 4

**Summary:**

The paper presents a method called ContraDiff, which leverages contrastive learning to guide an agent towards high-return states and pushing agents away from low-return states. The proposed method builds upon diffusion-based trajectory planning models and aims to solve offline reinforcement learning problems especially when high-return trajectories are limited in the offline dataset. Experiments are conducted on standard D4RL tasks and exhibit better performance as compared to baselines.

**Strengths:**

The combination of contrastive learning and diffusion-based trajectory planning model is a novel idea, allowing the model to learn from both high- and low-return samples. The main figure is clear and well-organized, making it easy to understand the method. Experiments are extensively conducted across many test environments.

**Weaknesses:**

1. While the idea of using contrastive learning is well motivated, the section on how "positive" and "negative" examples are identified requires additional explanations. The paper did not specify how the clustering is performed -- if the clustering is to enforce "dynamic consistency", I would assume the information is already available in the offline dataset? For me it is unclear how well the clustering captures reachability or state transitions beyond just grouping next states together. This could be a limitation in environments with complex or non-linear dynamics.
2. The paper lacks discussion on the complexity or runtime implications introduced by k-means clustering, especially for large offline RL datasets.
3. The increases in performance are mostly marginal. For many tasks the increase falls within one standard deviation of the baseline methods.
4. The authors mentioned  "comparing ContraDiff with other regular methods" but didn't include specific indication of where the results are presented. I later found the results in the appendix, but the authors may consider additional illustrations and explanations on the results. Furthermore methods like CQL are only tested on the standard D4RL datasets but not the mixture dataset proposed by the authors. More experiment results would be needed here to better demonstrate model performance.
5. It is unclear how the baseline results were obtained -- whether they came from existing code, were reimplemented by the authors, or taken from other sources.
6. The writing, especially in the experiments section, is not very clear to understand. Some sentences and typos make the paper harder to follow. For example:
* L290: "...high-return sample sparsity situations"
* L301: "...which are focus on addressing..."
* L364: "...declines with the ratio declines"
* Table3: "Walekr2d"

**Questions:**

Aside from mentioned in the Weakness section, my questions mostly come from experiments presented by the paper:
1. When comparing Table 3 with Table 1, it seems like introducing the expert trajectory leads to a decrease in performance, unexpectedly. For instance, DT achieves a score of 36.6 at original HalfCheetah-MR dataset but 7.5, 6.7, 6.1 in three conditions where expert data is introduced. I was wondering if the same codebase is used in achieving these results, and if so, could the authors share insights on why DT’s performance drops so drastically with the added expert data?
2. In most tasks ContraDiff-SR outperforms ContraDiff-SRD -- this seems unexpected given the author's intuition mentioned in section 3 that ContraDiff-SR may ignore the transition dynamics in its sampling process?
3. How would the proposed model perform on sparse-reward tasks like AntMaze? This benchmark is commonly used in offline RL evaluations but results for it are missing here.

---

> ### Author Response · Authors · 2024-11-21
> **Responses to Reviewer DdY7 (Q1-Q2)**
>
> **Q1.** While the idea of using contrastive learning is well motivated, the section on how "positive" and "negative" examples are identified requires additional explanations. The paper did not specify how the clustering is performed -- if the clustering is to enforce "dynamic consistency", I would assume the information is already available in the offline dataset? For me it is unclear how well the clustering captures reachability or state transitions beyond just grouping next states together. This could be a limitation in environments with complex or non-linear dynamics. \
> **R1.** Thank you for your comment.  We believe there is a misunderstanding regarding the details of clustering. \
> (1) We simply group all the states in the dataset into clusters. Specifically, for a state $s_t$, we first find the cluster it belongs to, marked as $C_t$.  Next, we treat all the next states of states in $C_t$ as the positive candidates of $s_t$. As all the states in $C_t$ are grouped into the same cluster as  $s_t$, we consider these states to be similar, and thus the next state of any in-cluster state of $C_t$ is also considered reachable from $s_t$. Thus, selecting the positive sample of $s_t$ from the next states of in-cluster states ensures the temporal consistency of the constructed contrastive samples. Note that all the required info is available in the dataset. \
> (2) However, as we understand it, you are referring to treating the same-cluster states of $s_t$'s next state as $s_t$'s next state. This represents a fundamental difference in exploration ability: In ContraDiff, the next states of same-cluster states are not necessarily within the same cluster, which allows ContraDiff to explore a broader range of potential next states. However, if we treat the same-cluster states of $s_t$'s next state as $s_t$'s next state since these states are all within the same cluster, they are inherently similar, which limits ContraDiff's exploration capacity.
> Overall, selecting contrastive samples from the next states of $s_t$'s same-cluster states is not the same as selecting contrastive samples from the same-cluster states of $s_t$'s next state. The next states of same-cluster states used in ContraDiff provide more diverse contrastive learning samples than the same-cluster states of $s_t$'s next state, and as a result, ContraDiff has greater exploration ability. \
> (3) We believe you are referring to scenarios where the environment dynamics may change over time. For these cases, we can use sub-segments of the trajectories as clustering units, and take into account both $s_t$​ and its historical information when searching for reachable future states in the training phase. Since historical information contains dynamics of the environment, clustering that incorporates historical information can effectively overcome the challenges posed by complex or non-linear dynamics. Exploration in non-linear dynamic scenarios is a promising direction, and we will consider it as part of our future work.
>
>
>
> **Q2.** The paper lacks discussion on the complexity or runtime implications introduced by k-means clustering, especially for large offline RL datasets. \
> **R2.** Thank you for your comment. Clustering is a preprocessing step, performed only once and stored in a file. During training, we only need to look up the clustering results, so clustering will not impact the training speed of the model. Moreover, the need for clustering information exists only during the training phase and is not required during the testing phase. \
> Nevertheless, we measured the time spent on clustering across different datasets, and the results are shown in the table below. Please note that we have taken several measures to optimize the clustering process. For example, we set the number of clusters to the square root of the number of states, and configure appropriate clustering batch sizes. It can be observed that even during the preprocessing stage, the clustering operation does not consume significant time.
>
> | Datasets                  | Number of States | Time Cost (Seconds) |
> |---------------------------|------------------|----------------------|
> | Walker2d-Medium-V2        | 1,000,000        | 116                  |
> | Walker2d-Medium-Replay-V2 | 302,000          | 75                   |
> | Walker2d-Random-V2        | 1,000,000        | 97                   |

---

> ### Author Response · Authors · 2024-11-21
> **Responses to Reviewer DdY7 (Q3-Q4)**
>
> **Q3.** The increases in performance are mostly marginal. For many tasks the increase falls within one standard deviation of the baseline methods. \
> **R3.** Thank you for your comment. In Section 4, we demonstrate the unique advantages of ContraDiff from multiple perspectives. Specifically, \
> （1）Table 1 presents the comparison between ContraDiff and several recent SOTA baselines on suboptimal datasets, where ContraDiff achieves the best or second-best performance on 25 out of 27 datasets. Even after accounting for the potential impact of variance, ContraDiff delivers outstanding results. For example, on HalfCheetah-Rand-Exp-0.1, ContraDiff outperforms the second-best method by 28.1%. This preliminarily demonstrates the advantage of ContraDiff in effectively leveraging negative samples. \
> （2）Furthermore, to better showcase the advantages of ContraDiff, we compare it with SOTA methods designed to handle imbalanced data. We initialize the environment with **random low-return states** to create more challenging scenarios, aiming to evaluate the ability of ContraDiff and SOTA methods to handle situations where the agent is trapped in low-return states. Specifically, we investigate the capability of ContraDiff to transition from low-return states to high-return states. As is shown in Table 2, ContraDiff achieves the best performance on almost all (23 out of 27) datasets compared to other SOTA baselines.
> For example, ContraDiff outperforms the second-best method by 41.4% on Hopper-Rand-Exp-0.1.
> Since the difference between ContraDiff and the baseline lies in leveraging negative sample information effectively through contrastive learning, results in Table 2 further highlight that utilizing low-return samples can effectively help the model escape from low-return areas. \
> （3）To provide a more intuitive explanation, we visualize in Figure 3 the ability of ContraDiff, AW+Diffuser, and their baseline, Diffuser, to escape from low-return states. As shown, when facing similar low-return states, AW+Diffuser achieves better results through its reweighting mechanism but still fails to fully escape the low-return region (terminating after 40 steps). In contrast, with the help of contrastive learning, ContraDiff effectively leverages negative sample information and maintains a healthy trajectory even after 60 steps, successfully escaping the low-return state. This visualization intuitively demonstrates the advantage of our proposed contrastive learning framework in fully utilizing negative sample information.
>
>
> **Q4.** The authors mentioned "comparing ContraDiff with other regular methods" but didn't include specific indication of where the results are presented. I later found the results in the appendix, but the authors may consider additional illustrations and explanations on the results. Furthermore methods like CQL are only tested on the standard D4RL datasets but not the mixture dataset proposed by the authors. More experiment results would be needed here to better demonstrate model performance. \
> **R4.** Thank you for your comment.
> （1）We have revised the description based on your comment to avoid potential misunderstandings. We summarize the results comparing ContraDiff with newly proposed SOTA methods in Table 1. However, due to the paper length limitation, we present the results compared with traditional methods in Table 3. The relevant analysis for comparisons with newly proposed SOTA methods is in Section 4.2.1, while the analysis for comparison with diffusion-free methods is in Appendix A.3. \
> (2) Nevertheless, based on your suggestion, we evaluated the performance of CQL, IQL, and MOPO on HalfCheetah-Rand-Exp, Walker2d-Rand-Exp, and Hopper-Rand-Exp, and compared them with ContraDiff. The results are shown in the table below. As can be observed, ContraDiff achieves the best performance in almost all scenarios.
>
> | Environment | Dataset    | Mix Ratio | CQL         | IQL         | MOPO        | ContraDiff  |
> |-------------|------------|-----------|-------------|-------------|-------------|-------------|
> | Halfcheetah | Rand-Exp   | 0.1       | 31.1 ± 1.6  | 3.9 ± 1.1   | 23.8 ± 2.1  | **48.0 ± 2.9**  |
> |             |            | 0.2       | 39.4 ± 3.2  | 70.4 ± 2.8  | 30.6 ± 1.1  | **72.3 ± 0.7**  |
> |             |            | 0.3       | 41.1 ± 0.9  | 72.7 ± 1.3  | 33.1 ± 1.9  | **88.7 ± 0.9**  |
> | Hopper      | Rand-Exp   | 0.1       | 9.2 ± 2.2   | 36.3 ± 2.9  | 12.9 ± 2.7  | **52.0 ± 0.7**  |
> |             |            | 0.2       | 20.8 ± 5.1  | 37.2 ± 0.9  | 17.0 ± 0.4  | **75.3 ± 1.0**  |
> |             |            | 0.3       | 50.7 ± 3.1  | 80.6 ± 1.1  | 36.8 ± 1.3  | **86.4 ± 1.5**  |
> | Walker2d    | Rand-Exp   | 0.1       | 11.6 ± 2.7  | **26.4 ± 2.1**  | 14.9 ± 2.4  | 20.2 ± 1.3  |
> |             |            | 0.2       | 13.9 ± 1.1  | 39.9 ± 1.7  | 30.7 ± 1.6  | **57.4 ± 0.7**  |
> |             |            | 0.3       | 34.2 ± 2.4  | 74.8 ± 2.2  | 44.9 ± 2.1  | **78.4 ± 1.2**  |

---

> ### Author Response · Authors · 2024-11-21
> **Responses to Reviewer DdY7 (Q5-Q9)**
>
> **Q5.**  It is unclear how the baseline results were obtained -- whether they came from existing code, were reimplemented by the authors, or taken from other sources. \
> **R5.** Thank you for your comment. The codes of other baselines were all obtained from their official repo. For the results of baselines in Table 1 and Table 2, we obtain the results by running the existing code of their official repo with default hyper-parameters.  The results of baseline methods in Table 3 are adopted from their own papers and Diffuser paper.
>
> **Q6.** The writing, especially in the experiments section, is not very clear to understand. Some sentences and typos make the paper harder to follow. \
> **R6.** Thank you for your comment! We have revised our paper based on your suggestions. Please refer to our revised paper for details.
>
> **Q7.** When comparing Table 3 with Table 1, it seems like introducing the expert trajectory leads to a decrease in performance, unexpectedly. For instance, DT achieves a score of 36.6 at original HalfCheetah-MR dataset but 7.5, 6.7, 6.1 in three conditions where expert data is introduced. I was wondering if the same codebase is used in achieving these results, and if so, could the authors share insights on why DT’s performance drops so drastically with the added expert data? \
> **R7.** Thank you for your comment. We have rechecked the code we used, and we confirm that the code we are using is from DT's official repo (as long as the other baselines). Note that since the DT paper demonstrated that the target return-to-go value has minimal impact on the model, we initially selected the maximum value from the dataset as the target return-to-go (i.e., RTG) for better performance. We pulled the code again from the official repository of DT and conducted relevant experiments, with results consistent with those reported in our paper.
>
> Still, we experimented with different return-to-go values on Halfcheetah-Rand-Exp, and the results are shown in the table below. It can be observed that excessively large target return-to-go values caused a performance decline. When the target return-to-go was reduced to 6000, we achieved comparable results. As DT achieves a score of 36.6 at the original HalfCheetah-MR dataset on the same setting, it can be concluded that expert data helps in improving DT's performance. Even though, ContraDiff still demonstrates a significantly better trend compared to DT.
>
> | Mix Ratio | DT (RTG=12000) | DT (RTG=6000) | $\text{ContraDiff}$ |
> |-----------|----------------|----------------|----------------------|
> | 0.1       | 7.5 ± 2.1      | 37.1 ± 0.9     | 39.0 ± 0.5          |
> | 0.2       | 6.7 ± 4.5      | 37.4 ± 0.8     | 58.4 ± 0.9          |
> | 0.3       | 6.1 ± 0.1      | 39.6 ± 2.2     | 60.3 ± 2.7          |
>
> **Q8.** In most tasks ContraDiff-SR outperforms ContraDiff-SRD -- this seems unexpected given the author's intuition mentioned in section 3 that ContraDiff-SR may ignore the transition dynamics in its sampling process? \
> **R8.** Thank you for your question. Ensuring temporal consistency is not always appropriate in all cases. In some environments and datasets, enforcing temporal consistency based on existing data may limit the model's exploration of states with higher returns. Although these states are not explicitly indicated as reachable in the dataset, they can be reached through the generalization capability of diffusion models, making ContraDiff-SR show better performance. Details of the scenarios where ensuring temporal consistency based on existing data can yield better results are discussed in Appendix A.4, please refer to our paper for details.
>
> **Q9.** How would the proposed model perform on sparse-reward tasks like AntMaze? This benchmark is commonly used in offline RL evaluations but results for it are missing here. \
> **R9.** Thank you for your suggestion! Following Diffuser and other baselines, we primarily evaluate ContraDiff's performance on locomotion, Maze2D, and Kitchen environments, in which Maze2D and Kitchen are sparse-reward tasks. Nevertheless, we have conducted experiments on Antmaze datasets, and the results are as follows. As can be observed, although Antmaze datasets are reward-diverse, ContraDiff outperforms Diffuser in most situations.
>
> | Datasets                     | Diffuser         | ContraDiff      |
> |------------------------------|------------------|-----------------|
> | Antmaze-umaze-v2             | 73.3 ± 3.2       | **78.0 ± 2.1**  |
> | Antmaze-umaze-diverse-v2     | 64.0 ± 1.1       | **68.5 ± 1.9**  |
> | Antmaze-large-play-v2        | 36.4 ± 1.3       | **72.7 ± 2.3**  |
> | Antmaze-large-diverse-v2     | 13.6 ± 0.7       | **20.0 ± 2.1**  |
> | Antmaze-medium-play-v2       | 45.0 ± 4.3       | **60.0 ± 0.9**  |
> | Antmaze-medium-diverse-v2    | **11.1 ± 2.7**   | 10.0 ± 1.1      |

---

> ### Author Response · Authors · 2024-11-25
> **Responses to Reviewer DdY7**
>
> Dear Reviewer DdY7,
>
> We have provided detailed responses to your concerns many days ago, we hope these responses have adequately addressed your concerns. As the discussion phase is coming to an end, we sincerely request your further responses.
>
> If we have resolved your issues, please consider raising your score to the positive side. If you have any further questions, please feel free to share them with us! Any valuable feedback is crucial for improving our work.
>
> Best regards, \
> The Authors

---

> > ### Comment · Reviewer_DdY7 · 2024-11-27
> > **Thank you for the detailed rebuttal**
> >
> > Thank you to the authors for the great efforts to address my concerns with the additional experiments! Most of my questions have been resolved. I have raised my score accordingly.

---

> > > ### Author Response · Authors · 2024-11-27
> > > **Responses to Reviewer DdY7**
> > >
> > > Dear Reviewer DdY7,
> > >
> > >
> > > Thank you for your effort in reviewing our paper, and thank you for acknowledging our work! Your valuable comments and suggestions have significantly contributed to improving our paper.
> > >
> > >
> > > Best regards, \
> > > The Authors

---

### Official Review · Reviewer_2zbW · 2024-11-04

**Soundness:** 2
**Presentation:** 3
**Contribution:** 2
**Rating:** 5
**Confidence:** 4

**Summary:**

This paper introduces ContraDiff, a novel offline reinforcement learning method that leverages contrastive learning to make better use of low-return trajectories by pulling states towards high-return regions and pushing them away from low-return areas. While the approach shows promising results across various environments and demonstrates advantages in handling sub-optimal datasets, several limitations remain, including unclear theoretical justification for the relationship between contrastive and RL losses, heavy reliance on hyperparameters for positive/negative sample selection, and lack of trajectory-level temporal information in the contrastive learning design. Despite these limitations, the paper presents a well-structured contribution with comprehensive experiments, providing a new perspective on utilizing sub-optimal data in offline RL.

**Strengths:**

1. The paper is well-structured and easy to follow, with clear motivation and comprehensive explanations of each component in the proposed method.

2. The empirical evaluation is thorough, covering 27 sub-optimal datasets and including extensive ablation studies, hyper-parameter analysis, and visualizations that help understand the method's behavior.

3. The proposed method provides a novel perspective on utilizing low-return trajectories in offline RL, and the implementation is relatively simple with only a few additional hyperparameters compared to the base diffusion model.

**Weaknesses:**

1. The relationship between contrastive loss and RL loss needs further detailed justification. Intuitively, these two losses appear highly correlated, especially when considering low-return trajectories as negative samples. A numerical analysis comparing these losses across trajectories with different returns would be valuable to verify if they indeed show similar trends. If so, the contrastive loss might merely serve as an enhancement to the RL loss rather than a meaningful regularization term that needs to be balanced against the primary objective.
2. The methodological differences illustrated in Figure 1(d)(e)(f) lack sufficient justification.This figure fails to demonstrate why pushing away from low-return states is fundamentally superior to upweighting high-return trajectories, or why return-based contrastive learning outperforms traditional trajectory-based approaches.
3. The paper's method of determining positive and negative samples relies heavily on hyperparameters (ξ, ζ, σ) without theoretical justification for their value ranges. The paper lacks a principled approach to determine these thresholds across different environments. A more systematic study on how different return thresholds affect the distribution of positive/negative samples and their impact on learning dynamics would strengthen the method's foundation.
4. The current state-level contrastive learning approach breaks the temporal correlation between states within trajectories, as it samples positive and negative states purely based on return values (SR) or clustering results (SRD). This design might lose important sequential information that could be better captured through trajectory-level contrastive learning. I encourage authors to explore trajectory-aligned state sampling strategies and conduct comparative experiments between state-level and trajectory-level contrastive learning to provide more solid empirical evidence for the design choices.
5. The competitive performance of baselines like CDE and HD raises questions about the necessity of the proposed contrastive mechanism, especially considering these methods achieve comparable results without any specific designs for handling low-return trajectories. While ContraDiff shows improvements in certain scenarios, the performance overlap with these simpler methods suggests that the advantages of explicitly handling low-return trajectories might be less significant than claimed. This observation calls for a more thorough analysis of what unique benefits, if any, are brought by the contrastive mechanism for low-return trajectory utilization.

**Questions:**

See weakness

---

> ### Author Response · Authors · 2024-11-21
> **Responses to Reviewer 2zbW  (Q1-Q2)**
>
> **Q1.** The relationship between contrastive loss and RL loss needs further detailed justification. Intuitively, these two losses appear highly correlated, especially when considering low-return trajectories as negative samples. A numerical analysis comparing these losses across trajectories with different returns would be valuable to verify if they indeed show similar trends. If so, the contrastive loss might merely serve as an enhancement to the RL loss rather than a meaningful regularization term that needs to be balanced against the primary objective. \
> **R1.** Thank you for your comments. \
> (1) The contrastive loss and RL loss are independent. The contrastive loss (i.e., Eq.(13) and Eq.(9)) is derived from InfoNCE, aiming to increase the expected return of generated trajectories by maximizing the distance from low-return trajectories while minimizing the distance to high-return trajectories. This effectively pulls the distribution of generated trajectories toward high-return trajectories and pushes it away from low-return trajectories. In contrast, the RL loss (i.e., Eq.(11)) simply attempts to fit the distribution of trajectories across the entire dataset.  \
> (2) Nevertheless, following your suggestion, we visualized the magnitude of the contrastive loss with and without contrastive learning on Walker2d-Rand-Exp-0.1, and presented this visualization in Appendix A.14 in our revised paper. Note that:
> $$
>     Eq.(13) = \\mathbb{E}_{t>0, i \sim [1, N]} [\\sum\_{h=t}^{t+H}\\frac{1}{h+1}\\mathcal{L}\_{h}^i  ],
> $$
> and
> $$
> \\mathcal{L}\_{h}^i =-\\log \\frac{ \\sum\_{k=0}^{\\kappa} \\exp( \\text{sim}( {f}(\\hat{s}\_h^{i,0}),  {f}({s}\_h^{+})  ) / T ) }{ \\sum\_{k=0}^{\\kappa} \\exp( \\text{sim}( {f}(\\hat{s}\_h^{i,0}), {f}({s}\_h^{-}) ) / T )  }.
> $$
> The closer the distance to high-return states and the farther the distance to low-return states, the smaller the value of Eq. (13). It can be observed that, regardless of the use of contrastive learning, the RL loss shows a downward trend and eventually stabilizes. However, for the value of Eq.(13), when contrastive learning is not used, the contrastive loss remains large and fluctuates significantly. On the other hand, when contrastive learning is applied, the contrastive loss decreases. Moreover, ContraDiff shows a lower value on RL Loss + Eq.(13) than ContrasDiff w/o CL.  This demonstrates that (1) the RL loss and contrastive loss are independent, with the contrastive loss serving as a meaningful regularization term; and (2) contrastive learning indeed maximizes the distance from low-return trajectories while minimizing the distance to high-return trajectories, as further corroborated by the analyses in Sections 4.3.
>
> **Q2.** The methodological differences illustrated in Figure 1(d)(e)(f) lack sufficient justification.This figure fails to demonstrate why pushing away from low-return states is fundamentally superior to upweighting high-return trajectories, or why return-based contrastive learning outperforms traditional trajectory-based approaches. \
> **R2.**  Thank you for your comment. As shown in Figure 1 (d), up-weighting high-return trajectories can only achieve the high-return trajectory ($s$ -> $G$) that exists within the dataset. However, when the agent is trapped in low-return states, e.g., $s_t$ in Figure 1 (d), there is no corresponding high-return example in the dataset (e.g., $s_t$ -> $G$), and up-weighting high-return trajectories fails. Our method, as shown in Figure 1 (e), by constraining the future states, takes the state from the high-return trajectory ($s$ -> $G$) as the target state at $s_t$, thereby enabling a transition from $s_t$ to a high-return trajectory. Besides, there are also some other methods that use contrastive learning in reinforcement learning (Figure (f)). However, they focus on learning better representations. For example, they aim to minimize the similarity between representations of adjacent states within the same trajectory while maximizing the distance between representations of states from different trajectories, as illustrated in Figure 1(f).

---

> ### Author Response · Authors · 2024-11-21
> **Responses to Reviewer 2zbW  (Q3-Q4)**
>
> **Q3.** The paper's method of determining positive and negative samples relies heavily on hyperparameters (ξ, ζ, σ) without theoretical justification for their value ranges. The paper lacks a principled approach to determine these thresholds across different environments. A more systematic study on how different return thresholds affect the distribution of positive/negative samples and their impact on learning dynamics would strengthen the method's foundation. \
> **R3.** Thank you for your comment. \
> (1) The main idea of our paper is to introduce contrastive learning to enhance decision-making. Therefore, we focus on evaluating the effectiveness of the idea instead of justifying the hyperparameters. As shown in Table 1 and Table 2, even using the most rudimentary positive and negative sample selection method (i.e., dividing solely based on thresholds), ContraDiff still demonstrates significant advantages in most scenarios.\
> As for the choice of threshold, a priori can be made based on the state return in the dataset. For example, one can visualize the state-return distribution of the dataset, and take the return with the most dramatic return change as the threshold. For example, thresholds can be identified as turning points by calculating the rate of change (e.g., differences) or the second-order rate of change (acceleration)[1,2]. After that, we can fix the threshold and adjust $\sigma$ for better results. \
> (2) We have conducted the experiments of thresholds for positive and negative samples and hyperparameters (ξ, ζ, σ)  in Appendix A.11.  In summary,  analysis of the hyper-parameters indicates that their impact on model performance is moderate, and searching for the optimal hyper-parameters is easy. Please refer to Appendix A.11 for more detailed discussions. \
> [1] Satopaa, V., Albrecht, J., Irwin, D., & Raghavan, B. (2011, June). Finding a" kneedle" in a haystack: Detecting knee points in system behavior. In 2011 31st international conference on distributed computing systems workshops (pp. 166-171). IEEE. \
> [2] Silverman, B. W. (2018). Density estimation for statistics and data analysis. Routledge.\
>
> **Q4.** The current state-level contrastive learning approach breaks the temporal correlation between states within trajectories, as it samples positive and negative states purely based on return values (SR) or clustering results (SRD). This design might lose important sequential information that could be better captured through trajectory-level contrastive learning. I encourage authors to explore trajectory-aligned state sampling strategies and conduct comparative experiments between state-level and trajectory-level contrastive learning to provide more solid empirical evidence for the design choices. \
> **R4.** Thank you for your comment and suggestion.
> (1) We believe by using "the important sequential information", you are referring to the dynamic consistency. We want to clarify that we designed the cluster-based approach (i.e., SRD) to ensure dynamic consistency, as is described in Section 3.2.1. Specifically, for a state $s_t$, we first find the cluster it belongs to, marked as $C_t$.  Next, we treat all the next states of states in $C_t$ as the positive candidates of $s_t$. As all the states in $C_t$ are grouped into the same cluster as  $s_t$, we consider these states to be similar, and thus the next state of any in-cluster state of $C_t$ is also considered reachable from $s_t$. Thus, selecting the positive sample of $s_t$ from the next states of in-cluster states ensures the temporal consistency of the constructed contrastive samples. \
> (2) Nevertheless, following your suggestion, we conducted experiments on trajectory-level contrastive learning (denoted as $\text{ContraDiff}^{\text{t}}$ ) on Walker2d-Rand-Exp, and the results are as follows. As can be observed, ContraDiff outperforms $\text{ContraDiff}^{\text{t}}$ in all scenarios, indicating that our state-level contrastive learning method does not significantly affect the sequential information.
>
>
> | Mix Ratio | $\text{ContraDiff}^{~\text{t}}$ | $\text{ContraDiff}$ |
> |-----------|--------------------------------|---------------------|
> | 0.1       | 19.2 ± 2.2                    | 20.2 ± 1.3          |
> | 0.2       | 27.7 ± 1.9                    | 57.4 ± 0.7          |
> | 0.3       | 45.1 ± 2.9                    | 78.4 ± 1.2          |

---

> ### Author Response · Authors · 2024-11-21
> **Responses to Reviewer 2zbW  (Q5)**
>
> **Q5.** The competitive performance of baselines like CDE and HD raises questions about the necessity of the proposed contrastive mechanism, especially considering these methods achieve comparable results without any specific designs for handling low-return trajectories. While ContraDiff shows improvements in certain scenarios, the performance overlap with these simpler methods suggests that the advantages of explicitly handling low-return trajectories might be less significant than claimed. This observation calls for a more thorough analysis of what unique benefits, if any, are brought by the contrastive mechanism for low-return trajectory utilization. \
> **R5.** Thank you for your comment. \
> （1）Table 1 presents the comparison between ContraDiff and several recent SOTA baselines on suboptimal datasets, where ContraDiff achieves the best or second-best performance on 25 out of 27 datasets. This preliminarily demonstrates the advantage of ContraDiff in effectively leveraging negative samples. \
> （2）Furthermore, to better showcase the advantages of ContraDiff, we compare it with SOTA methods designed to handle imbalanced data. We initialize the environment with **random low-return states** to create more challenging scenarios, aiming to evaluate the ability of ContraDiff and SOTA methods to handle situations where the agent is trapped in low-return states. Specifically, we investigate the capability of ContraDiff to transition from low-return states to high-return states. As is shown in Table 2, ContraDiff achieves the best performance on almost all (23 out of 27) datasets compared to other SOTA baselines. Since the difference between ContraDiff and the baseline lies in leveraging negative sample information effectively through contrastive learning, results in Table 2 further highlight that utilizing low-return samples can effectively help the model escape from low-return areas. \
> （3）To provide a more intuitive explanation, we visualize in Figure 3 the ability of ContraDiff, AW+Diffuser, and their baseline, Diffuser, to escape from low-return states. As shown, when facing similar low-return states, AW+Diffuser achieves better results through its reweighting mechanism but still fails to fully escape the low-return region (terminating after 40 steps). In contrast, with the help of contrastive learning, ContraDiff effectively leverages negative sample information and maintains a healthy trajectory even after 60 steps, successfully escaping the low-return state. This visualization intuitively demonstrates the advantage of our proposed contrastive learning framework in fully utilizing negative sample information.

---

> ### Author Response · Authors · 2024-11-25
> **Responses to Reviewer 2zbW**
>
> Dear Reviewer 2zbW,
>
> We have provided detailed responses to your concerns many days ago, we hope these responses have adequately addressed your concerns. As the discussion phase is coming to an end, we sincerely request your further responses.
>
> If we have resolved your issues, please consider raising your score to the positive side. If you have any further questions, please feel free to share them with us! Any valuable feedback is crucial for improving our work.
>
> Best regards, \
> The Authors

---

> ### Comment · Reviewer_2zbW · 2024-11-26
>
> Thank you for the authors' rebuttal. However, I regret that my fundamental concerns remain unaddressed:
>
> 1. The authors' characterization of RL loss as 'simply fitting trajectory distribution' is fundamentally incorrect and misleading. This description fits vanilla behavioral cloning, not RL losses, which are specifically designed to learn from high-return trajectories through value estimation and advantage weighting. The claimed independence between contrastive and RL losses is therefore questionable, as both inherently bias towards high-return trajectories.
>
> 2. The response merely differentiates the losses mathematically without demonstrating their fundamental independence or complementarity rather than redundancy.
>
> 3. Most critically, the authors fail to acknowledge or compare their work with 'Contrastive Energy Prediction for Exact Energy-Guided Diffusion Sampling in Offline RL', which presents a more theoretically complete treatment of combining contrastive learning with diffusion models in offline RL.
>
> Given these unresolved issues - the unclear relationship between contrastive and RL losses, insufficient comparison with critical baseline methods, and inadequate positioning relative to important related work - I believe this work does not yet meet the bar for publication.

---

> > ### Author Response · Authors · 2024-11-27
> > **Responses to Reviewer 2zbW (Q3)**
> >
> > **Q3.** Inadequate positioning relative to important related work "Contrastive Energy Prediction for Exact Energy-Guided Diffusion Sampling in Offline RL". \
> > **R3.** Thank you for your further comment.  Although QGPO [1] also employs contrastive learning, it addresses a different problem compared to ContraDiff. Generally speaking, QGPO [1] uses contrastive learning to learn a more accurate energy function as guidance, focusing on proposing a new energy-based guidance for Diffusion models. In contrast, as mentioned in R1, ContraDiff focuses on leveraging contrastive learning to address the data imbalance problem, so the two do not overlap. However, thank you for your suggestion, we have enriched our Related Works with QGPO. Please refer to our revised paper for details.
> >
> > [1] Lu, Cheng, et al. "Contrastive energy prediction for exact energy-guided diffusion sampling in offline reinforcement learning." International Conference on Machine Learning. PMLR, 2023.

---

> > ### Author Response · Authors · 2024-11-28
> > **Request for Further Discussion**
> >
> > Dear Reviewer 2zbW,
> >
> > We have provided detailed responses to your concerns days ago, and we have also fixed the issue in the previous response where formulas could not render properly. We hope our responses have adequately addressed your additional concerns. As the discussion phase is coming to an end, we sincerely request your further responses.
> >
> > If we have resolved your concerns, please consider raising your score to the positive side. If you have any further questions, please feel free to share with us! We would appreciate your further discussion on our paper.
> >
> > Best regards,
> > The Authors

---

> ### Author Response · Authors · 2024-11-27
> **Responses to Reviewer 2zbW (Q1 and Q2)**
>
> Dear reviewer, \
> We greatly appreciate you letting us know that some of your concerns have not been fully addressed.
> Also, we found that the formulas in our previous response (R1) were not rendered correctly. We sincerely apologize for any inconvenience this may have caused you. We have fixed the issue, and the formulas are now properly rendered.
> Here are our further responses:
>
>
> **Q1.**  The unclear relationship between contrastive and RL losses. The claimed independence between contrastive and RL losses is therefore questionable, as both inherently bias towards high-return trajectories. \
> **R1.** Thank you for your further comment. \
> Firstly, we want to clarify that the RL loss mentioned in our previous reply refers to the loss used in Diffusion-based methods such as Diffuser. In the works based on Diffusion for decision-making, such as Diffuser, they apply an MSE loss between real data and generated trajectories to train the Diffusion model. Only during evaluation do they adopt value-guided sampling to generate trajectories with higher returns. Therefore, the learned diffusion model simply fits the trajectory distribution. \
> However, the contrastive loss used in our paper is designed to better leverage the information embedded in negative samples (i.e., low-return trajectories). By maximizing the distance between generated trajectories and negative samples while minimizing the distance between generated trajectories and positive samples(i.e., high-return trajectories), the contrastive loss constrains the generated trajectories to a high-return area.
>
> Secondly, even methods based on the Actor-Critic framework, such as IQL and CQL, their loss basically assign weights to different ($s$, $a$) pairs based on the advantage of $a$, which fails in actively utilizing the information from low-return samples, for example,  they do not actively distance the generated samples from low-return trajectories. In contrast, by employing a contrastive loss, our model can fully leverage the information from low-return trajectories, and avoid low-return samples. The comparisons in Table 2 demonstrate the advantages of our proposed contrastive loss over these reweighting methods. \
>
> Furthermore, the difference between the contrastive loss and the RL loss can be visually demonstrated through the experiments.  Figure 5 and Figure 6 in Section 4.3 of our paper intuitively demonstrate the advantages of our contrastive loss. In Figure 5, we can observe that compared to without using contrastive loss (Figure 5(b)), employing contrastive loss (Figure 5(c)) results in more high-reward states, as indicated by the warmer-colored sample points in the figure. To further illustrate the advantages of the contrastive loss, Figure 6 presents the reward distributions when value guidance is not applied, comparing cases with and without contrastive loss. As shown, without contrastive loss (orange region), the rewards obtained by the model closely align with the reward distribution of the dataset itself (green region). However, when using contrastive loss (blue region), the reward distribution of the model is highly concentrated in the high-reward region, with minimal presence in the low-reward region. Figures 5 and 6 intuitively demonstrate the benefits brought by the proposed contrastive learning loss.
>
>
> **Q2.** Insufficient comparison with critical baseline methods. \
> **R2.** Thank you for your further comment.  \
> First of all, we want to clarify that, as mentioned in the Introduction, we focus on addressing the data imbalance problem by making better use of low-return samples in the dataset. Therefore, it is more fair to compare our method with other approaches aimed at solving data imbalance.
>
> As is shown in Table 2, we compare ContraDiff with data imbalance works that were recently published on ICLR 2023 and NIPS 2023, including advantage-weighting (AW) [1], return-weighting (RW) [1],  density-ratio weighting with advantage (AW-DW) [2] and density-ratio weighting with uniform (U-DW) [2]. The results show that ContraDiff achieves optimal or sub-optimal results in 25 out of 27 situations.  Besides, we have compared ContraDiff with plenty of regular SOTA RL methods in Table 1, in which ContraDiff achieves optimal or sub-optimal results in 25 out of 27 situations.
>
> [1] Hong, Zhang-Wei, et al. "Harnessing Mixed Offline Reinforcement Learning Datasets via Trajectory Weighting." The Eleventh International Conference on Learning Representations. \
> [2] Hong, Zhang-Wei, et al. "Beyond uniform sampling: Offline reinforcement learning with imbalanced datasets." Advances in Neural Information Processing Systems 36 (2023): 4985-5009.

---

> ### Author Response · Authors · 2024-12-01
> **Sincere Requests for Further Discussions**
>
> Dear Reviewer 2zbW,
>
> Thank you for your effort in reviewing our paper! We have provided detailed responses to your concerns several days ago, and we hope our responses have adequately addressed your additional concerns. As the discussion phase is coming to an end, we sincerely request your further responses.
>
> If we have resolved your concerns, please consider raising your score to the positive side. If you have any further questions, please feel free to share with us! We would appreciate your further discussion on our paper.
>
> Best regards,  \
> The Authors

---

> > ### Comment · Reviewer_2zbW · 2024-12-03
> >
> > Thank you for the detailed response. However, I remain unconvinced by your arguments for several fundamental reasons.
> >
> > 1.Your characterization of the relationship between RL and contrastive losses reveals concerning misunderstandings. The claim that RL loss in diffusion-based methods "simply fits trajectory distribution" is inaccurate - these methods already incorporate mechanisms to differentiate between high and low-return trajectories through value-guided sampling. Similarly, traditional actor-critic methods like IQL and CQL effectively handle trajectory relationships through value estimation and advantage weighting, inherently achieving a similar "pushing away" effect that your contrastive learning aims to accomplish. A deeper issue is that these approaches - Q/V values in actor-critic methods, value-guided sampling in diffusion models, and your proposed contrastive learning - are all fundamentally forms of trajectory guidance that bias learning towards high-return behaviors.
> >
> > 2.The claimed independence between RL loss and contrastive loss therefore requires much stronger theoretical justification. This connection has been well explored in works like QGPO, which leads to my second concern. **In the revision, your response that QGPO "addresses a different problem" is insufficient. This requires further comparison and discussion, as it is the work most closely related to this paper.** Both works use contrastive learning to improve diffusion-based offline RL, warranting performance comparisons on common benchmarks, detailed technical analysis of similarities and differences, and clear positioning of your contributions relative to QGPO. The current **one-line addition to Related Work** and lack of comparative analysis make it difficult to evaluate your work's novelty and contribution to the field. I believe these issues need to be thoroughly addressed to support your claims about the method's uniqueness and effectiveness.
> >
> > I believe this paper falls below acceptable standards on the several key issues, so I will maintain my score of 5.

---

> ### Author Response · Authors · 2024-12-04
> **Responses to Reviewer 2zbW (Q1 and Q2)**
>
> Dear Reviewer 2zbW, \
> Thank you for your further comment. \
> **Q1**. About the understanding of RL loss in diffusion-based methods. \
> **R1**. The diffusion model-based methods like Diffuser [1] are typically optimized with two losses: the diffusion loss and the guidance loss (i.e., the guidance you have mentioned). The diffusion loss is usually defined as MSE. For example, in Diffuser, the diffusion model loss is defined as (in page 4 of diffuser paper [1]):
>
> $$
> \\mathcal{L}(\\theta) = \\mathbb{E}\_{i,\\epsilon, \\tau^0}[ || \\epsilon - \\epsilon\_{\\theta}(\\tau^i, i) ||^2 ]
> $$
>
> The guidance loss is used to predict the value of each action,  which can be observed in the official repo of Diffuser. We agree with your opinion that previous methods adopt value-guided sampling can generate better actions (or trajectories). However, the capability of value-guided sampling is actually learned by the guidance loss instead of the diffusion loss.  As can be seen from the formula,
> return is not utilized in the diffusion loss, and therefore the diffusion loss essentially just **simply fits the trajectory distribution**.
>
> The effect of diffusion losses can be validated in our experimental results. In Figure 6, we visualize the probability density functions of the rewards obtained by ContraDiff and ContraDiff-C **when no guidance is used**, where ContraDiff-C denotes the setting of removing the contrastive learning loss from ContraDiff. It is evident that the probability density obtained by ContraDiff-C is closer to the distribution of the dataset than ContraDiff, indicating the diffusion loss essentially just **simply fits the trajectory distribution**. Moreover, ContraDiff is more concentrated in the high-reward region, indicating that our proposed contrastive learning indeed **actively moves away from low-return samples, and moves toward the high-return samples**.
>
> [1] Janner, Michael, et al. "Planning with diffusion for flexible behavior synthesis." arXiv preprint arXiv:2205.09991 (2022).
>
>
> **Q2.** Traditional actor-critic methods like IQL and CQL inherently achieve a similar "pushing away" effect that our contrastive learning aims to accomplish. \
> **R2.** Generally, the RL loss of advantage-based methods adopts a form similar to IQL, where the RL loss of IQL is as follows (Eq.7 from the IQL paper [1]）:
>
> $$
> \\begin{equation}
>     \\mathcal{L}\_{\\pi}(\\phi) = \\mathbb{E}\_{(s,a)\\sim \\mathcal{D}}[ exp( \\beta( Q\_{\\hat{\\theta}}(s,a) - V\_{\\psi}(s) ) )\\text{log}\pi\_{\\phi}(a|s) ]
> \\end{equation}
> $$
>
> As can be seen, for samples with low advantage, the weight is closer to 0, This means that when a sample has a very low return, these methods choose to neglect the information in those samples. Hence,  essentially speaking, these methods mainly focus on learning from higher-return samples (i.e., how to achieve a high return), and traditional actor-critic methods like IQL and CQL do not exhibit the 'pushing away' effect we mentioned. Ablation studies in Section 4.3.1 show that ignoring the information from low-return samples leads to poor performance in most cases.
>
>
> In contrast, our method learns from both low-return and high-return samples (i.e., how to achieve a high return and how to avoid the low-return), making our method able to **actively avoid the low-return area (our unique advantage)**.
> As shown in Figure 3, we initialize ContraDiff and the baseline methods in a low-return state to explore their ability to escape from the low-return state. As can be observed, even though AW, a method focused on learning from high-return samples, takes a few more steps than Diffuser, it eventually receives terminal signals. In contrast, ContraDiff goes further and ultimately maintains a healthy posture to keep moving forward. In other words, with the contrastive loss, ContraDiff makes it better to escape from the low-return region compared to methods based on advantage-weighting.
>
>
>
> Overall, **The key difference** between our method and previous works is previous methods focus on learning from high-return samples, but our method learns from both high and low-return samples.
>
>
> [1] Kostrikov, Ilya, Ashvin Nair, and Sergey Levine. "Offline reinforcement learning with implicit q-learning." arXiv preprint arXiv:2110.06169 (2021).

---

> ### Author Response · Authors · 2024-12-04
> **Responses to Reviewer 2zbW (Q3)**
>
> **Q3**. The difference between our method ContraDiff and QGPO [1]. \
> **R3**.  We are glad that we have reached a consensus on this point that, as we mentioned in **R3** of our last response, our method and QGPO utilize contrastive learning in **different ways** and are applied to solve **different problems**.
>
> Nevertheless, following your suggestion, we conducted experiments of QGPO on Walker2d-Rand-Exp, and the results are as follows. As can be observed, ContraDiff outperforms QGPO  in 2 out of 3 settings, indicating that ContraDiff,  leveraging contrastive learning to pull the generated samples closer to high-return samples and actively push them away from low-return samples, achieves better results compared to QGPO, which uses contrastive learning to learn higher guidance. We will also add the relevant experimental results to our paper.
>
> | Mix Ratio | QGPO | ContraDiff |
> |:--------:|:--------:|:--------:|
> | 0.1       | **22.5 ± 2.1** | 20.2 ± 1.3         |
> | 0.2       | 44.6 ± 1.9   | **57.4 ± 0.7**     |
> | 0.3       | 61.1 ± 3.3   | **78.4 ± 1.2**     |
>
>
> As we had included a section in Related Works (Section 5.3) that provides a detailed overview of methods using contrastive learning in RL, given the different approaches and objectives in utilizing contrastive learning, we have given an appropriate introduction to QGPO in Section 5.3.
>
> [1] Lu, Cheng, et al. "Contrastive energy prediction for exact energy-guided diffusion sampling in offline reinforcement learning." International Conference on Machine Learning. PMLR, 2023.

---

### Author Response · Authors · 2024-11-21
**General Response**

We would like to express our sincere appreciation to all reviewers for your constructive feedback. We have made the following adjustments and conducted additional experiments based on the your suggestions, all changes are marked in blue in the revised paper:
1. We have enhanced the descriptions of Figure 1.
2. We have enhanced the discussions of Related Works.
3. We have revised several sentences in our paper for better understanding, and have corrected the typos in our paper.

---

### Author Response · Authors · 2024-12-04
**Summary of Rebuttal**

Dear PC, SAC, AC and Reviewers,

We would like to express our sincere gratitude to all the reviewers for their efforts in reviewing. We are delighted that we have addressed the concerns of Reviewers Ku5d and DdY7, and they have increased their scores toward the positive side.

We also greatly appreciate the reviewers' recognition of our work. All the reviewers found our approach to be sufficiently novel and considered our experiments thorough and comprehensive. In particular, Reviewer Ku5d mentioned, "Instead of merely reweighting high-return samples, the method leverages low-return trajectories as guiding forces, creating a 'counteracting force' that helps the policy avoid suboptimal states." We believe this is an excellent summary of our work. With the introduction of the contrastive learning mechanism, ContraDiff is able to (1) fully leverage the information of both high and low-return samples, (2) actively escape from low-return regions.


We are also glad that we have addressed most of the concerns of Reviewer 2zbW. For unclosed questions of Reviewer 2zbW, we have provided a comprehensive analysis and additional experimental evidence. We believe our response can addresses the reviewer' concern.


We would like to once again express our sincere gratitude to all of the reviewers, the comments and suggestions from reviewers have been very pertinent and valuable, and we believe that this discussion has been highly efficient and fruitful.

Best Regards, \
The Authors

---

### Meta-Review · Area_Chair_XZQd · 2024-12-18

**Metareview:**

This paper proposes a method for offline RL that focuses on improving performance in regimes where there is a substantial proportion of data present from sub-optimal trajectories (a large ratio of low-return trajectories to high-return trajectories). The paper proposes a contrastive learning method to "constrain" (penalize in a loss function) the learned policy to high-return states and away from low-return states. The main claim is that the proposed method results in a performance improvement over comparable related work on sub-optimal datasets. The reviewers mostly agreed that the paper provided sufficient evidence to support this claim, although one reviewer pointed out a potentially relevant missing performance comparison (see additional comments below for more details). The authors responded by explaining the differences, including a brief mention of the differences in an updated submission, and providing a small amount of additional experiments that compared the results against the proposed work.

I concur with the reviewer that the mostly-missing performance comparison was plausibly relevant, and also observed that the preliminary experimental evidence that the authors followed up with provides a bit more evidence for the claim. I also found that the author's qualitative explanation of the difference plausible -- the related work is indeed applying contrastive learning to, at least on the surface, a different, (although potentially more general) purpose. I think this paper is acceptable, but would strongly benefit from a more comprehensive comparison on the remaining settings (the authors included additional experiments of the related work, QGPO, on 3/27 of the experimental settings in Table 2.) This evidence would provide a more complete performance comparison on the settings studied in the paper.

**Additional Comments On Reviewer Discussion:**

The main changes were primarily written clarifications, which addressed almost all of the main concerns.

The aforementioned remaining concern was raised by reviewer 2zbW:
>  In the revision, your response that QGPO "addresses a different problem" is insufficient. This requires further comparison and discussion, as it is the work most closely related to this paper. Both works use contrastive learning to improve diffusion-based offline RL, warranting performance comparisons on common benchmarks, detailed technical analysis of similarities and differences, and clear positioning of your contributions relative to QGPO. The current one-line addition to Related Work and lack of comparative analysis make it difficult to evaluate your work's novelty and contribution to the field. I believe these issues need to be thoroughly addressed to support your claims about the method's uniqueness and effectiveness.

The authors responded with the inclusion of 3 experiments using QGPO, illustrating that the proposed method outperforms it in 2/3 of the proposed regimes (the higher-ratio regimes of low-return data). The reviewer did not respond to these additional results. These experiments offer a bit of evidence that supports the authors' claim.

---

### Decision · Program_Chairs · 2025-01-22

Accept (Poster)